# High-temperature stability in air of $Ti_3C_2T_x$ MXene-based composite with extracted bentonite

Na Liu [1,2,3], Qiaoqiao Li [4], Hujie Wan [1,2], Libo Chang [1,2], Hao Wang[5], Jianhua Fang[3], Tianpeng Ding[1,2], Qiye Wen [1,2] ✉, Liujiang Zhou [2,4] ✉ & Xu Xiao [1,2] ✉

Although $Ti_3C_2T_x$ MXene is a promising material for many applications such as catalysis, energy storage, electromagnetic interference shielding due to its metallic conductivity and high processability, it's poor resistance to oxidation at high temperatures makes its application under harsh environments challenging. Here, we report an air-stable $Ti_3C_2T_x$ based composite with extracted bentonite (EB) nanosheets. In this case, oxygen molecules are shown to be preferentially adsorbed on EB. The saturated adsorption of oxygen on EB further inhibits more oxygen molecules to be adsorbed on the surface of $Ti_3C_2T_x$ due to the weakened $p$-$d$ orbital hybridization between adsorbed $O_2$ and $Ti_3C_2T_x$, which is induced by the $Ti_3C_2T_x$/EB interface coupling. As a result, the composite is capable of tolerating high annealing temperatures (above 400 °C for several hours) both in air or humid environment, indicating highly improved antioxidation properties in harsh condition. The above finding is shown to be independent on the termination ratio of $Ti_3C_2T_x$ obtained through different synthesis routes. Utilized as terahertz shielding materials, the composite retains its shielding ability after high-temperature treatment even up to 600 °C, while pristine $Ti_3C_2T_x$ is completely oxidized with no terahertz shielding ability. Joule heating and thermal cycling performance are also demonstrated.

MXenes are an emerging family of two-dimensional transition metal carbides and nitrides with a unit formula of $M_{n+1}X_nT_x$ ($n = 1–3$), where M represents an early transition metal (such as Sc, Ti, V, Cr, Mn, Y, Zr, Nb, Mo, Hf, Ta, et al.), X corresponds to nitrogen or carbon, $T_x$ stands for the surface functional groups (such as -OH, -F, =O)[1–3]. By virtue of the high conductivity and solution processability, MXenes show great potential in Joule heating, energy conversion/storage, electromagnetic interference (EMI) shielding, etc. For instance, compared to the conventional dense metallic films and graphene with hydrophobic surface[4,5], self-assembled ultrathin MXene film (a 24-layer film of ≈55 nm thickness) is capable of providing 99% reflection shielding (EMI SE of 20 dB) in microwave band[6]. Notably, besides the traditional microwave band, MXenes are demonstrated as highly efficient shielding or absorbing material for higher-frequency terahertz waves (THz, frequency ranging from 0.1 to 10 THz), which is assigned for 6 G wireless communication and has many potential applications in aerospace industry[7–10]. However,

[1]School of Electronic Science and Engineering, State Key Laboratory of Electronic Thin Film and Integrated Devices, University of Electronic Science and Technology of China, Chengdu, Sichuan 611731, China. [2]Yangtze Delta Region Institute (Huzhou), University of Electronic Science and Technology of China, Huzhou, Zhejiang 313001, China. [3]Department of Petroleum, Oil and Lubricants, Army Logistic Academy of PLA, Chongqing 401331, China. [4]School of Physics, University of Electronic Science and Technology of China, Chengdu, Sichuan 611731, China. [5]Research Institute of Superconductor Electronics, School of Electronic Science and Engineering, Nanjing University, Nanjing 210023, China. ✉e-mail: qywen@uestc.edu.cn; ljzhou86@uestc.edu.cn; xuxiao@uestc.edu.cn

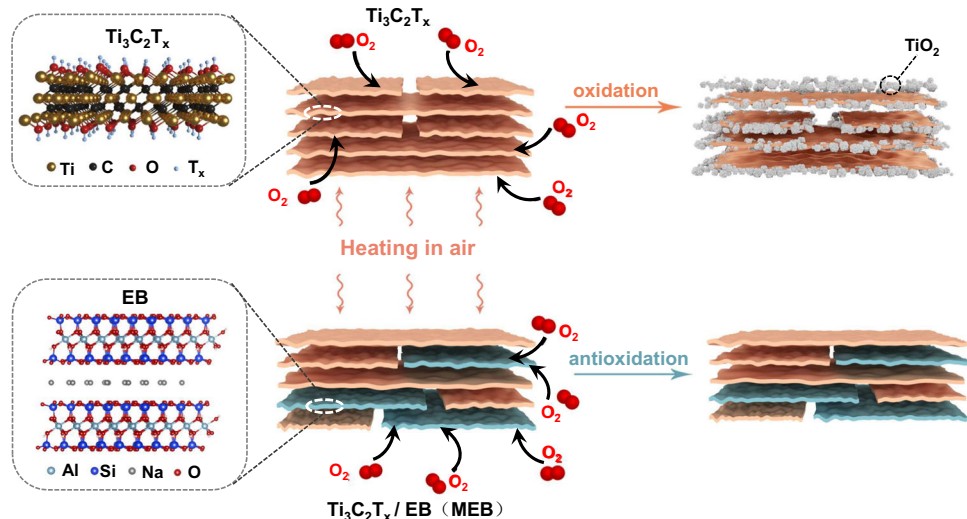

**Fig. 1 | Schematic shows the oxidation process of pristine Ti3C2Tx (starting from the edge) and suppressed oxidation of MEB under high-temperature annealing with the presence of oxygen.** The top row shows the oxidation of pristine $Ti_3C_2T_x$, which is induced by interacting with oxygen molecules, and $Ti_3C_2T_x$ finally transforms to $TiO_2$. The bottom row shows the suppressed oxidation of MEB, which is induced by the prior adsorption of oxygen molecules on EB compared to $Ti_3C_2T_x$. Even after saturated adsorption of $O_2$ on EB, further $O_2$ molecules are still repelled onto the surface of $Ti_3C_2T_x$ due to the coupling between $Ti_3C_2T_x$ and EB.

the good performance of MXenes is always achieved at room temperature or low-temperature due to their nature of poor antioxidation properties in air and humid environment at elevated temperature, attributed to thermodynamically metastable surface[11]. This may greatly limit their further application towards practical high-temperature scenarios (for instance, a heater is required to work under hundreds of degrees in some cases, EMI shielding of engine casting of aircraft needs to endure high temperatures above 380 °C[12], etc.).

Previously, the oxidation mechanisms of MXenes, especially for $Ti_3C_2T_x$, have been investigated[13–17]. In general, both oxygen and water molecules are responsible for the oxidation of the $Ti_3C_2T_x$, even $Ti_3C_2T_x$ preserved in degassed water would be oxidized after a while[14]. This oxidation process would be accelerated at elevated temperatures with the presence of oxygen or water molecules[18]. Accordingly, numerous efforts have been focusing on the antioxidation of $Ti_3C_2T_x$, mainly by eliminating $O_2$ and/or $H_2O$ from contacting $Ti_3C_2T_x$, which could be summarized into three strategies. First, changing the storage condition of $Ti_3C_2T_x$ (such as ultralow freezing temperature[19], inert or reductive atmosphere[13,20], organic solvents[21]) is an effective way to protect the flakes from oxidation. Second, applying antioxidants (such as ascorbate[22] and inorganic salts[23]) can prolong the storage lifetime of $Ti_3C_2T_x$ MXene in solutions. Third, a homogeneous surface encapsulation strategy has been used to isolate $Ti_3C_2T_x$ from $O_2$ and/ or $H_2O$[24,25]. However, the above strategies mainly concentrate on the antioxidation of $Ti_3C_2T_x$ in an aqueous solution or room temperature. The high-temperature resistant property of $Ti_3C_2T_x$ in the air or humid environment for a long operation time remains unexplored while highly important.

Herein, we report an air-stable $Ti_3C_2T_x$ MXene-based composite that could endure high-temperature annealing in the air. Restacked film composed of pristine $Ti_3C_2T_x$ flakes with metallic conductivity up to 24,000 S cm$^{-1}$ is a good candidate for applications requiring high conductivity. However, the oxidation process of $Ti_3C_2T_x$ into $TiO_2$ with the presence of oxygen under high temperature largely decreases its conductivity, as shown in Fig. 1. To solve this issue, we propose a competition mechanism by compositing $Ti_3C_2T_x$ with extracted bentonite (EB). EB is a layered material consisting of one Al octahedral sheet sandwiched between two Si tetrahedral nanosheets with Na$^+$ ions to offset the charge imbalance[26] (bottom left corner in Fig. 1). Interestingly, EB shows stronger $O_2$ adsorption compared to $Ti_3C_2T_x$. Even

after saturated adsorption of $O_2$ on EB, further $O_2$ molecules are still repelled onto the surface of $Ti_3C_2T_x$ due to the coupling between $Ti_3C_2T_x$ and EB. In this context, $Ti_3C_2T_x$/EB (MEB) composite could endure annealing at above 400 °C for several hours in an air and humid environment.

## Results and discussion
### Fabrication and characterization of MEB
$Ti_3C_2T_x$ was obtained by selectively etching the $Ti_3AlC_2$ MAX phase (see details in Methods). EB was achieved by extracting and delaminating sodium bentonite powder (see details in Methods). The successful synthesis of those two materials was confirmed by X-ray diffraction (XRD) patterns (Fig. S1), scanning electron microscopy (SEM), and transmission electron microscopy (TEM) images (Fig. S2), which show typical nanosheet morphology with lateral size of ~8 µm for $Ti_3C_2T_x$ and 2–5 µm for EB, respectively[1,26,27]. As shown in Fig. S3a, the mixed aqueous dispersions of $Ti_3C_2T_x$/EB are aggregated when the mass percentage of EB is more than 50 wt%. With the Zeta potential of −42 mV (Fig. S3b) close to sole $Ti_3C_2T_x$ or EB, the homogeneously mixed dispersion (Fig. S3c) with an EB mass ratio of 50 wt% is chosen to fabricate $Ti_3C_2T_x$/EB composite film for the following study and be denoted as MEB. MEB maintains a similar lamellar structure and flexibility compared to the restacked $Ti_3C_2T_x$ and EB films (Fig. S4), which is further confirmed by the images of scanning transmission electron microscopy (STEM). As shown in Fig. 2a, layered and homogeneous distributions of Si and Ti elements indicate the stacking of EB and $Ti_3C_2T_x$.

As mentioned previously, $O_2$ contributes to the oxidation of $Ti_3C_2T_x$ in the air, which would be accelerated at elevated temperatures. Therefore, we take the annealing time and temperature as variables to investigate the antioxidation of MEB under synthetic air (with volume fractions of 21% $O_2$ and 79% $N_2$). The samples treated at different conditions are denoted as sample-air-T-t, where the sample is $Ti_3C_2T_x$, EB, or MEB, T is annealing temperature, and t is annealing time. The pristine $Ti_3C_2T_x$ and MEB at room temperature are the control samples denoted as $Ti_3C_2T_x$-RT and MEB-RT.

As shown in Fig. 2b, MEB-Air-400C-2 reserves the flexibility of fresh MEB and keeps its integrity after ultrasonication for 1 min in DI water (Fig. S5b). By contrast, $Ti_3C_2T_x$-Air-400C-2 tends to be fragile (Fig. 2b and Fig. S5a). According to the tensile stress-strain test (Fig. 2c and Fig. S6), MEB-Air-400C-2 exhibits a 10.6% decrease in tensile stress

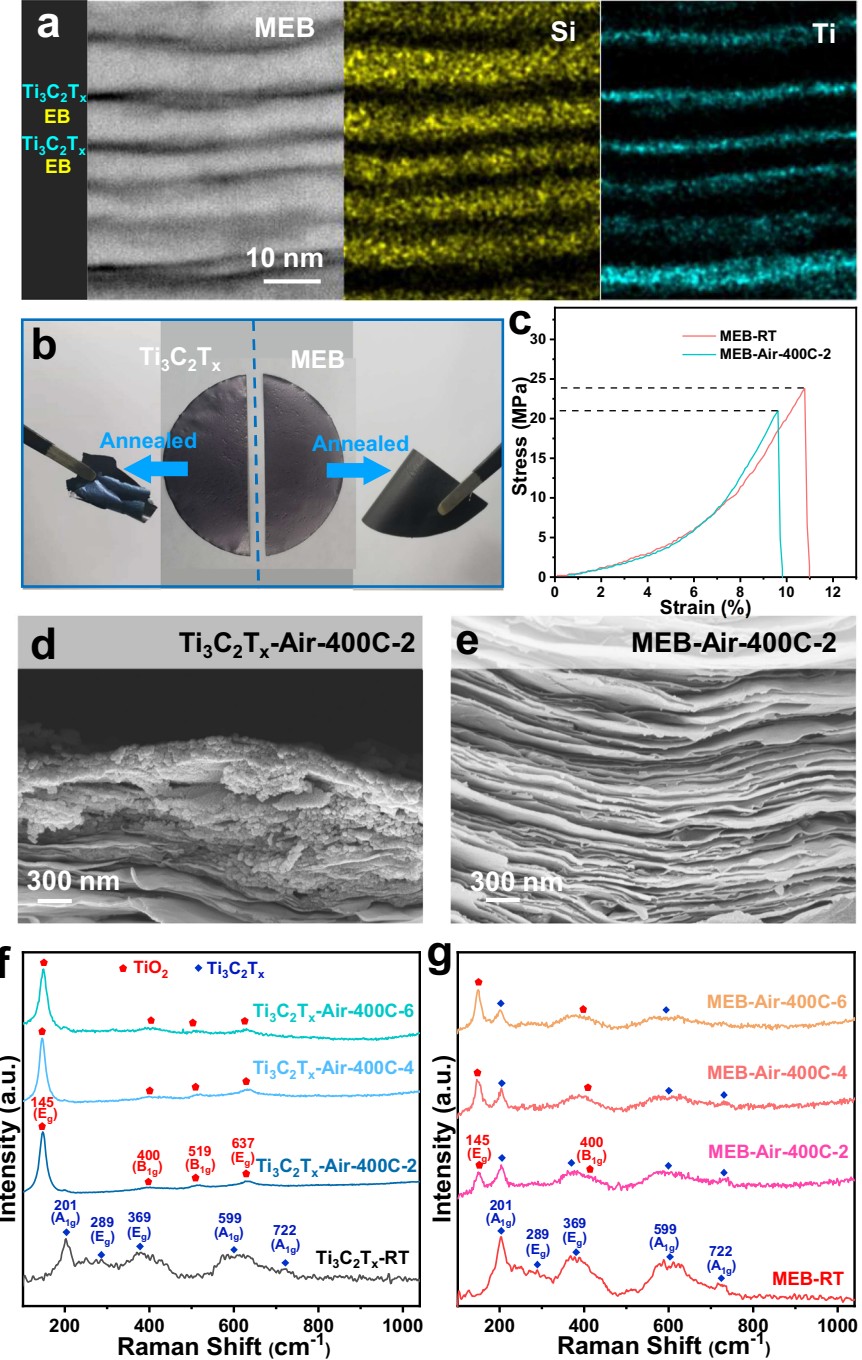

**Fig. 2 | Characterization of pristine Ti3C2Tx and MEB under high-temperature annealing with the presence of oxygen. a** Cross-sectional STEM images of MEB with EDX mapping. **b** Optical image of pristine Ti$_3$C$_2$T$_x$ film and MEB film before and after annealing at 400 °C for 2 h in synthetic air. **c** The tensile stress-strain curves of MEB-RT and MEB-Air-400C-2. Cross-sectional SEM images of the annealed Ti$_3$C$_2$T$_x$ films (**d**) and MEB films (**e**). **f** Raman spectra of Ti$_3$C$_2$T$_x$ films before and after treatment under synthetic air at 400 °C for 2, 4, and 6 h. **g** Raman spectra of MEB films before and after treatment under synthetic air at 400 °C for 2, 4, and 6 h. Source data are provided as a Source Data file.

and a 1.0% decrease in strain. By contrast, Ti$_3$C$_2$T$_x$-Air-400C-2 exhibits a 57.7% decrease in tensile stress and a 5.5% decrease in strain, which is caused by the degradation of Ti$_3$C$_2$T$_x$ in high-temperature treatment. The cross-sectional SEM images (Fig. 2d, e) show good retention of layered structure for MEB-Air-400C-2 and partial transformation from nanosheets to particles in Ti$_3$C$_2$T$_x$-Air-400C-2, demonstrating that the presence of EB nanosheets can delay the oxidation of Ti$_3$C$_2$T$_x$. Further measurements are utilized to investigate the high-temperature stability of MEB under the atmosphere of synthetic air. Raman spectra (Fig. 2f, g) display that typical E$_g$ (Ti, C) mode at 289, 369 cm$^{-1}$ and A$_{1g}$

(Ti, C) mode at 201, 599, and 722 cm$^{-1}$ of Ti$_3$C$_2$T$_x$[28,29] remain high strength in MEB-Air-400C-2 and A$_{1g}$ peak (201 cm$^{-1}$) is still obvious in MEB-Air-400C-6. In sharp contrast, no typical peaks of Ti$_3$C$_2$T$_x$ remain, while the peaks of TiO$_2$ at 145, 400, 519, and 637 cm$^{-1}$ appear in Ti$_3$C$_2$T$_x$-Air-400C-2, showing the transformation of Ti-C bonding to Ti-O bonding to the great extent[30,31]. Interestingly, the morphology of MEB remains the lamellar structure even after treatment at 400 °C for 6 h in the air, as shown in cross-sectional SEM images (Fig. S7c), while pristine Ti$_3$C$_2$T$_x$ flakes transform into thicker chunk under the same treatment condition (Fig. S8c). The crystalline structure analyses of annealed

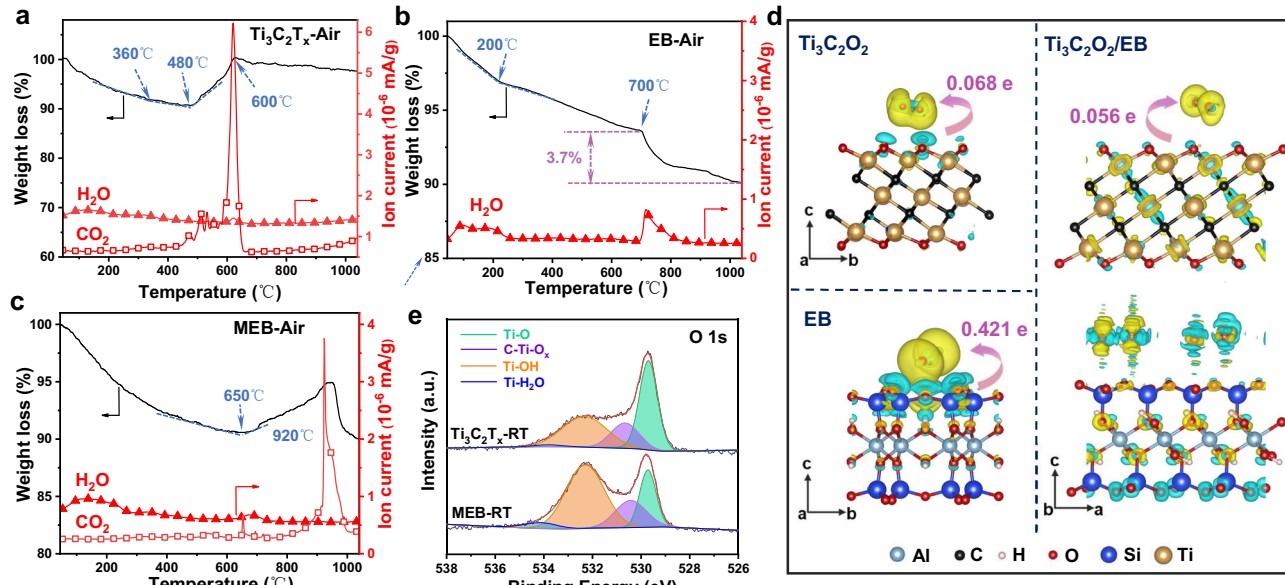

**Fig. 3 | Mechanism of high-temperature resistance in MEB.** Thermal gravimetric (TG) curves in the air with mass spectrometry analysis (MS) for the atomic mass unit (amu) of 18/$H_2O$ and 44/$CO_2$ for (**a**) $Ti_3C_2T_x$, (**b**) EB, and **c** MEB. **d** The charge density difference plots for the stable configurations of one $O_2$ adsorbed on $Ti_3C_2O_2$, EB, and $Ti_3C_2O_2$/EB heterostructure. In the heterostructure, EB is subject to saturated oxygen adsorption. The isosurface level is set to be 0.0002 e/Å³ except for $O_2$ adsorbed on EB with a value of 0.0006 e/Å³. The yellow area indicates charge accumulation, and the green region represents charge depletion. **e** X-ray photo-electron spectroscopy (XPS) of O 1s of $Ti_3C_2T_x$-RT and MEB-RT. Source data are provided as a Source Data file.

MEB (Fig. S9b) and $Ti_3C_2T_x$ (Fig. S9a) further confirm the suppression of oxidation for MEB at high temperatures in the air, even for a long treatment time.

**Mechanism of high-temperature resistant property in MEB**

The mechanism of high-temperature resistant property in MEB is investigated experimentally and theoretically. The thermogravimetry-mass spectrum (TG – MS) was used to reveal detailed insight into the thermal stability of $Ti_3C_2T_x$, EB, and MEB in synthetic air or argon (Ar). As shown in Fig. 3a, $Ti_3C_2T_x$ exhibits four stages of weight change at 100, 360, 480, and 600 °C, respectively. At 100 °C, the removal of intercalated water leads to the appearance of the first $H_2O$ peak and weight loss. The second weak $H_2O$ peak is observed at 360 °C due to the slow dissociation of hydroxyl surface terminations[32]. The weight loss at 100–360 °C is in agreement with the TG – MS of $Ti_3C_2T_x$ in Ar (Fig. S10a), which indicates that the degradation of $Ti_3C_2T_x$ in synthetic air starts at ~360 °C. This is also the reason why we set the temperature variable above 400 °C in this work. Around 480 °C, the weight of $Ti_3C_2T_x$ quickly increases due to the interaction with $O_2$, which evolves into $TiO_2$ accompanied by the generation of $CO_2$[33]. The oxidation process accelerates with the increase of annealing temperature and approaches completion at 600 °C. For EB, as shown in Fig. 3b and S10b, only $H_2O$ is detected from the mass spectrum in the heating process under air and Ar atmospheres, suggesting the high thermal stability of EB with further confirmed by XRD and SEM analyses in Figs. S11–S13. The degradation of EB starts at 700 °C because of the dissociation of -OH and =O terminations[26]. Notably, EB exhibits a weight loss of 3.7% at 700–1040 °C in air, larger than that in Ar (2.5%). We assume that EB may interplay with $O_2$, thereby inducing the change in weight loss during the $O_2$ adsorption and desorption. This is evident that the rate of weight loss of EB in synthetic air changes at around 200 °C (Fig. 3b), while a negligible change in the rate of weight loss can be observed from the TG curve in Ar (Fig. S10b). Interestingly, as shown in Fig. 3c, MEB exhibits much better resistance to oxidation than pure $Ti_3C_2T_x$, as demonstrated by (i) the weight of MEB starts to increase at 650 °C with the appearance of $CO_2$, which is much higher than pure $Ti_3C_2T_x$ of 480 °C; (ii) $Ti_3C_2T_x$ in MEB transforms into $TiO_2$

completely at 920 °C with the strongest signal of $CO_2$, while 600 °C is enough for $Ti_3C_2T_x$. Since EB is thermally stable in synthetic air during annealing, and the interaction between the surface of $Ti_3C_2T_x$ and $O_2$ plays a key role in the oxidation process, we reasonably speculate that the induction of EB may affect the interaction between adsorbed $O_2$ and $Ti_3C_2T_x$ by the $Ti_3C_2T_x$/EB interface coupling.

To further reveal the mechanism of the high-temperature resistant property of MEB, theoretical calculations were performed based on density functional theory (DFT). The adsorption energy ($E_{ad}$) and Bader charge states of $O_2$ molecule adsorbed on $Ti_3C_2O_2$ and EB substrates are listed in Table 1. $Ti_3C_2O_2$ was chosen as the prototype model due to the O-termination is the dominant way in experimentally synthesized $Ti_3C_2T_x$ MXene (as shown in Fig. 4d). The more negative $E_{ad}$ for the oxygen adsorbed on EB indicates the stronger coupling when compared with that on $Ti_3C_2O_2$. Meanwhile, the Bader charge analysis[34] was performed in order to quantitatively assess the amount of charge transfer within the adsorbed systems, as shown in Fig. 3d and S14a, b. The adsorbed $O_2$ obtains 0.421 electrons on EB, much larger than that on $Ti_3C_2O_2$ (0.068 electrons), further confirming the stronger binding ability between $O_2$ and EB.

Besides, we calculated the $E_{ad}$ of oxygen adsorbed on $Ti_3C_2O_2$ that is interfaced with EB (denoted as $Ti_3C_2O_2$/EB heterojunction). Considering that EB would firstly couple with oxygen, four $O_2$ molecules were firstly placed on the surface of EB to mimic the saturated adsorption of $O_2$ and then EB was interfaced with $Ti_3C_2O_2$ in the

**Table 1 | The calculated adsorption energy $E_{ad}$ and Bader charge state that transferred from substrates to adsorbed $O_2$ molecules**

| Materials | $E_{ad}$ (eV) | Bader charge (electron) |
|---|---|---|
| EB | −0.916 | 0.421 |
| $Ti_3C_2O_2$ | −0.254 | 0.068 |
| $Ti_3C_2O_2$/EB | −0.126 | 0.056 |
| $h$-BN | −0.143 | 0.055 |

The positive (negative) sign of $E_{ad}$ indicates the adsorption of the $O_2$ molecule is energetically unfavorable (favorable).

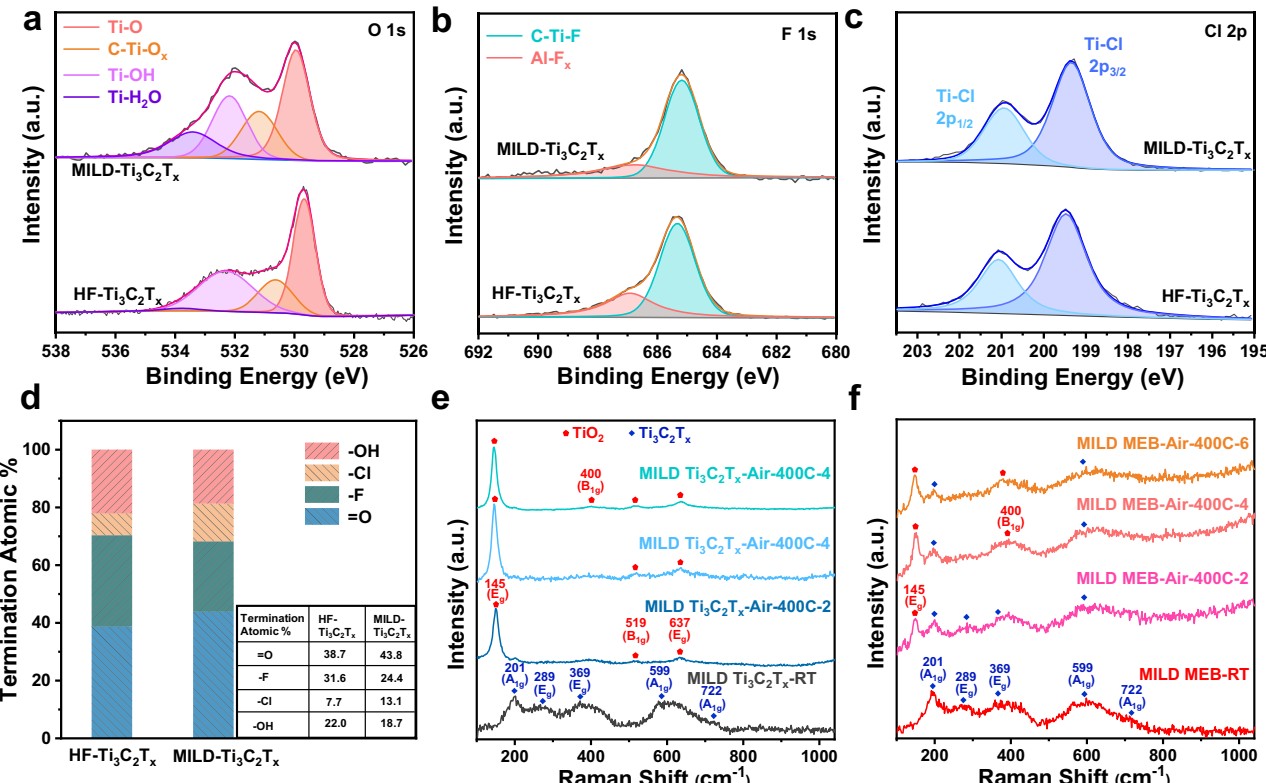

**Fig. 4 | Different termination ratios of MILD-Ti3C2T$x$ and HF-Ti3C2T$x$, and high-temperature resistant property of MILD MEB with the presence of oxygen. a–c** XPS of O 1$s$, F 1$s$, and Cl 2$p$ signals, respectively, for the pristine HF-Ti$_3$C$_2$T$_x$ and MILD-Ti$_3$C$_2$T$_x$ films. **d** The termination ratio of HF-Ti$_3$C$_2$T$_x$ and MILD-Ti$_3$C$_2$T$_x$ was measured by XPS with a table list (inserted). **e** Raman spectra of MILD Ti$_3$C$_2$T$_x$ before and after treatment under synthetic air at 400 °C for 2, 4, and 6 h. **f** Raman spectra of MILD MEB before and after treatment under synthetic air at 400 °C for 2, 4, and 6 h. Source data are provided as a Source Data file.

heterojunction model (Fig. S14c). The $E_{ad}$ of one O$_2$ molecule on Ti$_3$C$_2$O$_2$/EB heterojunction (Fig. 3d) is −0.126 eV, which is less negative than that of pure Ti$_3$C$_2$O$_2$ (−0.254 eV) (Table 1). This indicates a significant decrease in the interaction of Ti$_3$C$_2$O$_2$ with oxygen in Ti$_3$C$_2$O$_2$/EB heterojunction. The Bader charges of adsorbed O$_2$ on heterostructure is 0.056 electrons, smaller than that on pure Ti$_3$C$_2$O$_2$ (0.068 electrons), in line with the charge density difference plot (Fig. 3d). In addition, the charge density difference plot shows that the adsorbed O$_2$ on freestanding Ti$_3$C$_2$O$_2$ mainly obtains electrons from the nearest-neighbor Ti-O sub-layer of Ti$_3$C$_2$O$_2$. Upon Ti$_3$C$_2$O$_2$ is interfaced with EB (O$_2$ molecules saturatedly adsorbed), the existence of the interface produces a smooth channel for charge transfer through the whole heterostructure along the $c$ direction (Fig. S14c). This leads to a charge accumulation around the saturated O$_2$ layer (four O$_2$ molecules) and a deficient domain of electron states in the upper Ti$_3$C$_2$O$_2$ layer. When one more O$_2$ is adsorbed on the outermost Ti$_3$C$_2$O$_2$ layer of Ti$_3$C$_2$O$_2$/EB heterostructure, the amount of electron loss in the intermediate region is greatly reduced, thus weakening the coupling of O$_2$ with the Ti$_3$C$_2$O$_2$ layer. This is associated with the weakened hybridization of O-$p$ orbitals of adsorbed O$_2$ molecules and Ti-$d$ orbitals. All of these indicate that the formation of the Ti$_3$C$_2$O$_2$/EB interface greatly inhibits the further adsorption of O$_2$ and thus enhances environmental stability.

The F-terminated MXene is also calculated as F termination accounts for a high proportion of T$_x$ (Fig. 4d). The calculated $E_{ad}$ and Bader charge states of Ti$_3$C$_2$F$_2$ before and after adsorption of O$_2$ show that Ti$_3$C$_2$F$_2$ exhibits more inferior interaction with O$_2$ compared to EB (Fig. S15), suggestive of similar behavior to that of Ti$_3$C$_2$O$_2$.

To further confirm the proposed mechanism, a 2D $h$-BN nanosheet with high thermal stability[35] (Fig. S16) is chosen to composite with Ti$_3$C$_2$T$_x$. Theoretically, $h$-BN shows inferior adsorption of O$_2$ to Ti$_3$C$_2$O$_2$, which is proved by higher negative $E_{ad}$ values (−0.143 eV) and less

amount of Bader charge transferred (0.055 electrons) of $h$-BN (Fig. S17 and Table 1). The Ti$_3$C$_2$T$_x$/$h$-BN composite was prepared by mixing two materials with a mass ratio of 1/1 (denoted as MBN). After annealing at 400 °C for 2 h in synthetic air, the layered structure of MBN transforms into a nanoparticle@nanosheet structure, as shown in Fig. S18. The peaks of TiO$_2$ at 145 cm$^{-1}$ appear with high intensity in the Raman spectrum (Fig. S19), indicating the oxidation of MBN. Since both of $h$-BN and EB exhibit good heat retardant property, the different thermal stability performance of their composites with Ti$_3$C$_2$T$_x$ excludes the possibility that the high-temperature resistant property of MEB comes from the heat retardant property of EB. Actually, as MEB was treated at a high temperature for more than 2 h, the heat distribution around/in MEB should be highly uniform. In this context, we believe the high-temperature resistant property of MEB in the air should be attributed to the superior O$_2$ adsorption on EB and the coupling between EB and Ti$_3$C$_2$T$_x$ that largely weakens the further adsorption of O$_2$ on Ti$_3$C$_2$T$_x$.

In addition, we found another phenomenon that may be beneficial for a high-temperature resistant property of MEB. As shown in X-ray photoelectron spectroscopy (XPS, Fig. 3e and S20), compared to pure Ti$_3$C$_2$T$_x$ (44% Ti-O, 35% Ti-OH), the ratio of Ti-O species reduced to 21% while Ti-OH species increased to 55% in MEB. It reveals that EB with abundant hydroxy (confirmed by Fourier transform infrared spectroscopy in Fig. S21) interacts with Ti-O terminations of Ti$_3$C$_2$T$_x$ to form hydrogen-bond, inducing compact layered structure as confirmed by SEM (Fig. S4b) and XRD analyses with the shift of (002) peak from 2θ = 5.8° (Ti$_3$C$_2$T$_x$) to 6.0° (MEB) (Fig. S9a, b). This is further evidenced by the higher permeability of O$_2$/H$_2$O of Ti$_3$C$_2$T$_x$ than MEB, as shown in Table S1, which reveals the better barrier property of MEB possibly benefited by the compact layer structure. As a result, the MEB composite could suppress the diffusion of O$_2$ to some extent, lowering the amount of O$_2$/H$_2$O around Ti$_3$C$_2$T$_x$.

As we mentioned, $H_2O$ is another factor for oxidizing $Ti_3C_2T_x$ at elevated temperatures. Here, we also study the high-temperature resistant property of MEB in the $H_2O$ atmosphere (argon with 90% relative humidity, denoted as RH 90%). The Raman spectra in Fig. S22 show that the intensities of $Ti_3C_2T_x$ peaks in $Ti_3C_2T_x$-RH 90%-400C-2 become weak. Further prolonging of annealing time results in more severe oxidation which can be confirmed by the appearance of $TiO_2$ $E_g$ peak at 145 cm$^{-1}$ in pure $Ti_3C_2T_x$ film after annealing for 4 h. However, negligible change of $Ti_3C_2T_x$ fingerprints is detected in MEB-RH 90%-400C-4. Along with the results of SEM (Figs. S23, S24) and XRD (Fig. S9c, d), it verifies the good thermal stability of MEB in the $H_2O$ atmosphere. In addition, the antioxidant capability of MEB is detected at higher temperatures (500 and 600 °C) under RH 60% synthetic air (simulated atmospheric environment). The layered structure of MEB retains after annealing for 2 h at 600 °C; on the contrary, the layered $Ti_3C_2T_x$ transforms into amorphous $TiO_2$ completely, which is in agreement with the XRD analysis (Fig. S25). All the results confirm that MEB demonstrates high-temperature resistant property even under long-time treatment with the presence of both oxygen and water molecules.

## The effect of termination ratio on the high-temperature resistant property of MEB

To detect the influence of terminations on the high-temperature resistant property of $Ti_3C_2T_x$, two approaches are adopted to synthesize $Ti_3C_2T_x$, which are minimally intensive layer delamination method (denoted to MILD-$Ti_3C_2T_x$, Fig. S26) and HF etching method (HF-$Ti_3C_2T_x$), respectively. By fitting the relative intensities of the O 1$s$, F 1$s$, Cl 2$p$ peaks[36,37] as shown in Fig. 4a–d and S27, we determine the termination of HF-$Ti_3C_2T_x$ to be 38.7% = O, 31.6% -F, 7.7% -Cl, 22.0% -OH and MILD-$Ti_3C_2T_x$ with 43.8% = O, 24.4% -F, 13.1% -Cl, 18.7% -OH, which reveals the different termination ratios for two samples. For clear description, a composite consisting of MILD-$Ti_3C_2T_x$ and EB is denoted as MILD MEB in order to differentiate with MEB (HF-$Ti_3C_2T_x$/EB). Following the same thermal treatment of MEB, MILD MEB-Air-400C-2 shows high-intensive $E_g$ (Ti, C) peaks at 289, 369 cm$^{-1}$ and $A_{1g}$ (Ti, C) peaks at 201, 599, 722 cm$^{-1}$ of $Ti_3C_2T_x$, and $A_{1g}$ peak (201 cm$^{-1}$) is still obvious in MILD MEB-Air-400C-6 as shown in Fig. 4f. By contrast, MILD $Ti_3C_2T_x$-Air-400C-2 (Fig. 4e) shows the transformation of Ti-C bonding to Ti-O bonding, which is indicated by the appearance of $TiO_2$ peaks at 145, 400, 519, 637 cm$^{-1}$. Along with the SEM analysis (Fig. S28), the results confirm the high-temperature resistant property for MILD MEB in air. The similar behavior in MEB and MILD MEB reveals our strategy for suppressing the oxidation of $Ti_3C_2T_x$ is independent of the termination ratio of $Ti_3C_2T_x$.

## The potential applications of MEB

High-temperature resistant electromagnetic interference (EMI) shielding is of importance for the aerospace industry, for scenarios including but not limited to the EMI shielding of engine casing of aircraft (higher than 380 °C), etc. Here we tested the EMI shielding efficiency of MEB in the frequency range of 0.2–1.3 THz after annealing at different temperatures for a long time in RH 60% synthetic air (simulated atmospheric environment in a practical scenario), which are denoted as MEB-Atmos-T-t (T is temperature, t is time) for different annealing conditions. Pristine $Ti_3C_2T_x$ is taken for comparison. In order to truly reflect the performance comparison, MEB and $Ti_3C_2T_x$ are controlled to obtain the same amount of 11 mg $Ti_3C_2T_x$, as the EMI shielding originates from the high electrical conductivity of $Ti_3C_2T_x$ and EB is noneffective for THz shielding (Fig. S29), which is further explained by the proposed EMI shielding mechanism of MEB in Fig. S30. As shown in Fig. S31, the transmitted THz signals of $Ti_3C_2T_x$-RT and MEB-RT are too weak to be detected due to the excellent THz shielding performance of $Ti_3C_2T_x$, and the average THz shielding efficiencies (THz SEs) are obtained around 47 dB as shown in Fig. 5a–c. After annealing, MEB-Atmos-400C-6 have a good shielding capacity retention, even higher than pristine

material (~50 dB). This can be attributed to the increase of the electrical conductivity from 850 to 1000 S cm$^{-1}$, which is caused by more compact nanosheet stacking of MEB-Atmos-400C-6 with the removal of intercalated water. The XRD analysis validates the reduction of the interlayer ($d$-) spacing, characterized by the peak of (002) shifting from 2θ = 6.0° to -8.2° after annealing (Fig. S9b). By contrast, $Ti_3C_2T_x$-Atmos-400C-6 performs high transmittance and ultralow THz SE (approach to zero) owing to its complete degradation. Besides, THz transmittance and shielding efficiency of MEB after annealing at a higher temperature for 2 h are investigated, as shown in Fig. 5b, c. With the annealing temperature increasing, the transmissions of THz waves through MEB-Atmos-500C-2 and MEB-Atmos-600C-2 remain at 0.001 and 0.0015%. In other words, the THz SE of the two samples can reach about 50 dB at 500 °C and 48 dB at 600 °C (Fig. 5b, c) because of good electric conductivity (1090 S cm$^{-1}$ for MEB-Atmos-500C-2h and 960 S cm$^{-1}$ for MEB-Atmos-600C-2h), which is consistent with the XRD analysis (Fig. S25d). On the contrary, THz SE of $Ti_3C_2T_x$ decreases to ~ 0 dB at 600 °C because of the oxidation of $Ti_3C_2T_x$. All the results suggest that the introduction of EB can suppress the oxidation-induced deterioration of $Ti_3C_2T_x$, therefore making MEB promising high-temperature resistant THz shielding materials. In addition, a comprehensive literature review is summarized to compare the performance of MEB with published THz shielding materials. So far, most THz shielding materials are utilized at room temperature and few researchers focus on their applications at high temperature or hash environment, as shown in Fig. S32. Therefore, MEB with high-temperature resistant property in air and humid environment may meet the demands of working in a harsh environment (further high-temperature THz test is necessary to finally evaluate the feasibility of MEB for the practical scenario).

To further explore the potential application of MEB by virtue of the high-temperature resistant property, we also measure the Joule heating performance of MEB. Joule heating device is required to provide fast heating and high-temperature output (in some cases, hundreds of degrees are needed). In general, the metallic conductivity and high thermal conductivity of $Ti_3C_2T_x$ meet the demand for the Joule heating device. The $Ti_3C_2T_x$ and MEB freestanding films were used to make the devices of the Joule heater (Fig. S33). As shown in Fig. 5d, pristine $Ti_3C_2T_x$ film quickly reaches 190 °C under a driving voltage of 3.0 V. However, it is stable for only 0.57 h, then the temperature begins to decline due to the oxidation of $Ti_3C_2T_x$. As a comparison, the heating rate of MEB film can be 20 °C s$^{-1}$, with a steady-state temperature of 198 °C without large temperature fluctuation for more than 3.5 h, which reflects the rapid thermal response and stable high-temperature electrothermal performance. In addition, the thermal cycling performance of MEB is tested as shown in Fig. 5e. The steady-state temperatures of MEB under driving voltage being on/off is basically identical during 30 cycles, which reveals the stable thermal cycling of MEB. The above results suggest that the capabilities of fast thermal response and thermal cycling stability make MEB potentially suitable for thermal/photothermal catalysis and beyond. In addition, a comparison of the steady temperature and heating rate for various Joule heating materials is listed in Table S3.

In conclusion, we present a $Ti_3C_2T_x$-based composite that is capable of suppressing oxidation at high temperatures (higher than 400 °C) in air. We demonstrate that oxygen is preferentially adsorbed on EB due to superior oxygen adsorption. The saturated adsorption of oxygen on EB further inhibits more oxygen molecules to be absorbed on the surface of $Ti_3C_2T_x$ due to the weakened $p$-$d$ orbital hybridization between adsorbed $O_2$ and $Ti_3C_2T_x$ that is induced by the $Ti_3C_2T_x$/EB interface coupling. In addition, our strategy for suppressing the oxidation of $Ti_3C_2T_x$ is independent of the termination ratio of $Ti_3C_2T_x$. Utilized as THz shielding material, MEB shows a high THz EMI SE of 48 dB after annealing under 600 °C in an atmospheric environment, which validates MEB as a promising candidate for THz shielding material that works in a high-temperature scenario. Joule heating and

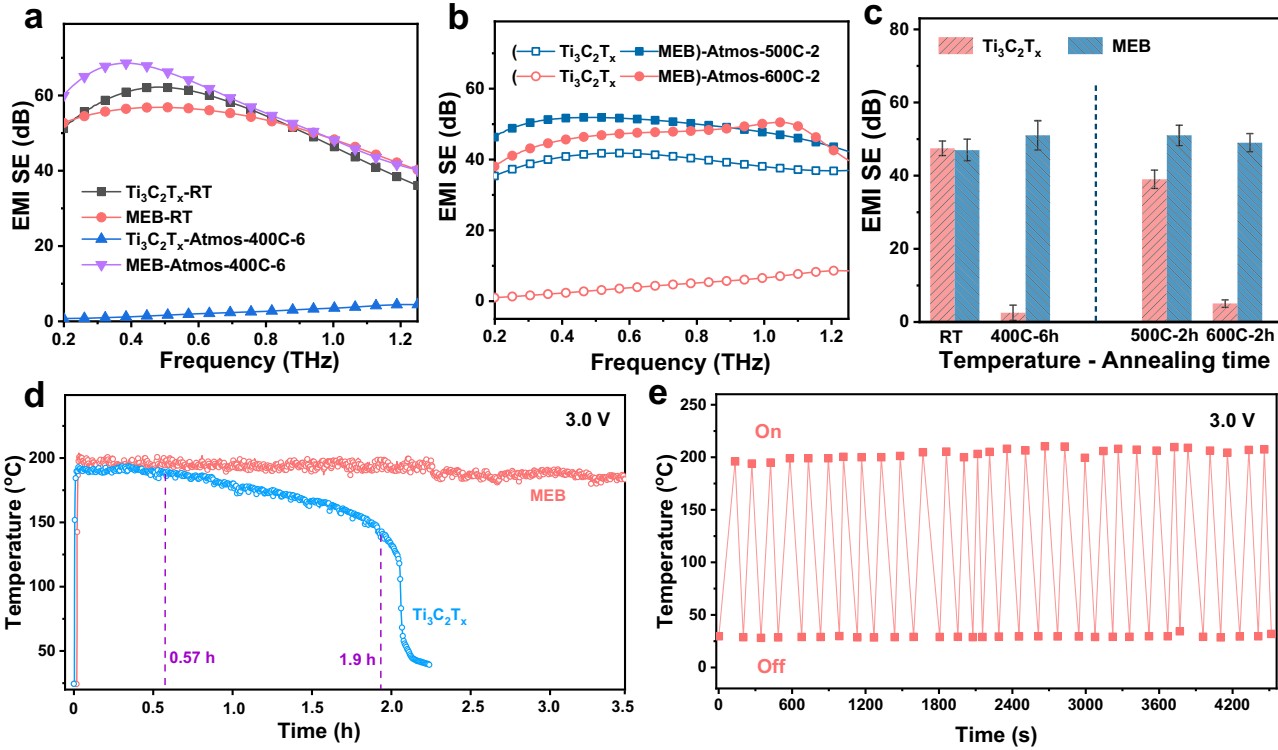

**Fig. 5 | Potential applications of MEB. a–c** THz EMI shielding property of MEB. **a** EMI SE in 0.2–1.3 THz of $Ti_3C_2T_x$-RT, MEB-RT, $Ti_3C_2T_x$-Atmos-400C-6, MEB-Atmos-400C-6. **b** EMI SE in 0.2–1.3 THz of $Ti_3C_2T_x$-Atmos-500C-2, 600C-2, and MEB-Atmos-500C-2, 600C-2. **c** The average THz SE of the samples showed in Fig. 5a (left) and Fig. 5b (right). **d** Long-term Joule heating performance of MEB and $Ti_3C_2T_x$ films driven by an applied voltage of 3.0 V. **e** The thermal cycling performance of MEB film, which is realized by switching the applied voltage (3.0 V) on and off repeatedly. Source data are provided as a Source Data file.

thermal cycling properties are also tested to expand the potential applications. The strategy of designing high-temperature resistant MXene-based composite may be applicable to other two-dimensional materials and beyond.

## Methods

### Synthesis of pristine $Ti_3C_2T_x$ and EB

we prepared $Ti_3C_2T_x$ by two approaches in this study. ***HF-$Ti_3C_2T_x$*** (denoted as $Ti_3C_2T_x$ in the paper for readability): $Ti_3C_2T_x$ was synthesized by selectively etching Al of $Ti_3AlC_2$ MAX (characterized by SEM and XRD[38], Fig. S34, S35) using HF and HCl[10,39]. About 1 g of $Ti_3AlC_2$ powder (11 Technology Co.) was added to the etchant solution, which includes 12 mL 38 wt% hydrochloric acid (HCl, Sigma-Aldrich), 2 mL 49 wt% hydrofluoric acid (HF, Sigma-Aldrich), and 6 mL deionized (DI) water. Next, the $Ti_3AlC_2$ powder was etched for 24 h by string at 400 rpm at 35 °C. Then, the multilayer $Ti_3C_2T_x$ was obtained by washing the reaction mixture with DI water via centrifugation at 684.78 × g (5 min per cycle). After washing 5–6 times (until the pH ≥ 6), the sediment was dispersed into the delamination solution, which was prepared by adding 1 g of LiCl (Sigma-Aldrich) into 40 mL DI water. Then, the mixture was stirred for 4 h at room temperature. The dispersion was washed using DI water and the single layer $Ti_3C_2T_x$ aqueous solution was obtained by centrifugation at 684.78 × g for 5 min. ***MILD-$Ti_3C_2T_x$*** (prepared by minimally intensive layer delamination method): LiF (1.6 g; Sigma-Aldrich) was added to 9 M HCl (20 mL; HCl, 38 wt%, Sigma-Aldrich) and continuously stirred at 400 rpm for 30 min at 40 °C. About 1 g $Ti_3AlC_2$ (11 Technology Co.) was added to the mixture solution and stirred at 400 rpm for 24 h at 40 °C. Then the dispersion was washed with DI water 4–6 times until the pH ≥ 6. After washing, the sediment was dispersed in 50 mL of fresh DI water and the single layer $Ti_3C_2T_x$ aqueous solution was obtained by centrifugation at 684.78×g for 5 min[40].

To prepare the EB aqueous solution, sodium bentonite powder (San Ding Technology Co.) was extracted and delaminated. The pristine powder was dissolved in DI water. Then, the homogeneous EB supernatant was collected after ultrasonic treatment for 3 h and centrifugation at 684.78 × g for 5 min. The concentrations of $Ti_3C_2T_x$ and EB dispersions were measured by filtering a specific amount of colloidal solution through a polypropylene filter (3501 Coated PP, Celgard LLC, Charlotte, NC), followed by drying under vacuum at 60 °C overnight and measuring the weight of the solid residue.

### Preparation of freestanding $Ti_3C_2T_x$ film, EB film, and MEB

About 11 mg of freestanding pristine $Ti_3C_2T_x$ film (with a thickness of ~5 μm) and 22 mg of freestanding pristine EB film (with a thickness of 11.2 μm) were fabricated by vacuum filtration of aqueous solutions, then the films were dried under vacuum at 60 °C overnight. The MEB solutions (with 11 mg $Ti_3C_2T_x$ and 11 mg EB) were prepared by adding EB aqueous solution into the $Ti_3C_2T_x$ solution and continued stirring for 2 h. The obtained homogeneous solution was filtrated by vacuum-assisted filtration and dried under vacuum at 60 °C for 24 h to obtain MEB (with a thickness of 11 μm).

Meanwhile, MILD $Ti_3C_2T_x$ film and MILD MEB are fabricated by MILD $Ti_3C_2T_x$ solution in a similar process.

### Preparation of freestanding MBN

*h*-BN nanosheets were prepared by delaminating *h*-BN powder (Aladdin) through ultrasonication in DI water for above 8 hours[35]. Then, the dispersion was centrifugated at 684.78×g for 5 min to collect the homogeneous layered *h*-BN supernatant and the concentration was measured by the same method of $Ti_3C_2T_x$ supernatant. MBN film was prepared by vacuum-assisted filtration of $Ti_3C_2T_x$/BN mixture, in which the weight percentage of BN was 0.5.

## Annealing treatment of materials

The sample was put into a quartz boat, then annealed in a tube furnace (OTF-1200X, Kojing Material Technology Co.) under a specific atmosphere (synthetic air, RH 90% argon, or simulated atmospheric environment) with a programmed heating rate of 5 °C min⁻¹. During the annealing process, the bottom surface of the sample may interact with less oxygen/water molecules than the upper surface due to the contact between the bottom surface of the sample and the quartz boat. This may result in less oxidation of the bottom surface than the top layer.

## Material characterization

Scanning electron microscopy (SEM) images were acquired at 15 kV by using a JSM-7600F SEM. X-ray diffraction (XRD) was performed on a Bruke D8 advanced powder X-ray diffractometer equipped with Copper Kα radiation ($\lambda = 1.540598$ Å). Transmission electron microscope (TEM) images were obtained from Tecnai G2 F20 S-TWIN, FEI (200 kV). Raman spectra were acquired using HORIBA's Raman system with a 532 nm excitation laser at 5% laser power (Jobin Yvon S.A.S, France). XPS was conducted using PHI VersaProbe 5000 instrument (Physical Electronics) with a 100 μm and 25 W monochromatic Al-Kα (1486.6 eV) X-ray source. Thermal gravimetric with mass spectrometry analysis (TG-MS) curves were obtained under a synthetic air/Ar atmosphere using STA449F3 and QMS 403 (Netzsch, Germany) with a mass spectrometer (110/220 V) and a temperature scan rate of 5 °C min⁻¹ from room temperature to 1100 °C. Electrical conductivities were tested by an ST 2253 four-probe test instrument (Suzhou Jingge Electronic Co., Ltd.). The samples for cross-sectional scanning transmission electron microscopy (STEM) were prepared by an FEI HELIOS NanoLab 600i Focused Ion Beam (FIB) system and the local element distribution (mapping) was analyzed by highly efficient energy dispersive X-ray (EDX) spectroscopy at 200 kV by FEI Talos F200X. Mechanical tests of films were performed on a dynamic mechanical analyzer WDW-10D and DS2-50N-XB (Baichuan Instruments Co.). The $Ti_3C_2T_x$ film samples were cut into 10 mm × 30 mm strips using a steel razor blade. Uniaxial tensile tests were performed at a strain rate of 5 mm/min. The sample thickness was estimated by the helical micrometer. The $O_2$ permeability was measured using a permeability tester (OX-TRAN 2/12 R, MOCON Co., USA). The water vapor transmission rate was measured using a moisture permeability testing apparatus (C360, Jinan Languang Electromechanical Technology Co.).

## Computational details

The theoretical calculations were performed by density functional theory (DFT)[41], as implemented in PWmat code[42,43]. The norm-conserving pseudopotentials (NCPP-SG15)[44] with Perdew–Burke–Ernzerhof (PBE) exchange-correlation functional were adopted[45]. The slab model of EB was built based on its (001) surface. A vacuum layer of 20 Å was utilized in the z direction to eliminate the coupling between the substrate sheet and its periodic replica. 2 × 2, 3 × 3, 3 × 3, and 4 × 4 supercells of (001)-plane EB, $Ti_3C_2O_2$, $Ti_3C_2F_2$ MXenes, and BN monolayer were utilized, respectively, so as to guarantee the close surface area in xy plane. Moreover, a $2\sqrt{3} \times 3$ supercell of (001)-plane MXene was interfaced with a 2 × 2 supercell of (001)-plane EB (denoted as $Ti_3C_2O_2$/EB) to model the inhibiting effect of $O_2$ adsorbed on MXene[34]. The lattice mismatch of the heterojunction is only about 1.5%. The plane-wave basis set cutoff energy was set to 60 Ry, and all atoms within systems were fully relaxed until the force on each atom was less than 0.02 eV/Å. The PBE-D3 method of Grimme was adopted to correct the long-range van der Waals interaction in all the calculations[46].

The adsorption energy $E_{ad}$ were quantitatively determined by the following equation:

$$E_{ad} = E_{(Slab+M)} - E_{Slab} - E_M \tag{1}$$

Where $E_{(Slab+M)}$ is the total energy of the M molecule adsorbed on the slab. $E_{Slab}$ and $E_M$ correspond to the energy of the slab model and M molecule, respectively.

## Terahertz shielding measurements

Terahertz time-domain spectroscopy (THz-TDS) was measured by titanium sapphire with a center wavelength of 800 nm and a pulse width <50 fs as the excitation source[9,10]. The measurements were performed at room temperature. The humidity in the test room was controlled in a low range (<5%). The effective spectrum range is 0.2–1.3 THz, and the repetition frequency is 1 kHz. The testing time step was about 0.067 ps throughout the experiment. The EMI SE values were calculated using the following equation:

$$EMI\,SE = 10\log(1/T) = -20\log(E_t/E_a) \tag{2}$$

Where T is the transmittance of terahertz wave; $E_t$ and $E_a$ refer to the amplitudes of transmission terahertz pulses for the samples and the air cavity, respectively.

## Reporting summary

Further information on research design is available in the Nature Research Reporting Summary linked to this article.

## Data availability

Source data are provided with this paper.

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

## Acknowledgements

This work is supported by the Foundation of National Natural Science Foundation of China (62235004 and 61831012), Outstanding Scholarship Foundation of UESTC (A1098531023601243), and the Sichuan Science and Technology Support Program (2021JDTD0026). N.L. would like to thank Mr. Yao Li for the helpful discussion at the initial stage.

## Author contributions

X.X. conceived the idea. N.L., Q.L., Hu.W., and L.C. carried out the theoretical calculation and experiments. N.L., Q.L., Hu.W., L.C., Q.W., L.Z., T.D., and X.X. analyzed the data. Ha.W. and J.F. assisted in the analysis. N.L., Q.L., Hu.W., L.C., Ha.W., Q.W., L.Z., and X.X. wrote the manuscript. All of the authors commented on the manuscript.

## Competing interests

The authors declare no competing interests.
