## [Peer Review File · Nature Communications]

High-temperature stability in air of $Ti_3C_2T_x$ MXene-based composite with extracted bentoniteReviewers' comments:

Reviewer #1 (Remarks to the Author):

Authors have reported an air-stable Ti₃C₂T_x based composite (MEB) by compositing Ti₃C₂T_x with extracted bentonite (EB) nanosheets. Based on their research study, air-stable Ti₃C₂T_x MXene based composite could endure high-temperature annealing in air or humid environment. Even though, the idea behind this research study is interesting but design of experiments is done poorly and does not meet the requirements for the scientific research study. Objectives should be clear in line with adapted methodology. The adapted methodology for the combination of MXene and EB should clarify the orientation of layers clearly as the intercalation play important role for the sake of reduction in oxidation factor. Figure 1 which reflects the orientation of layers for the synthesized composite should be proven clearly with strong support as this mainly affect the purpose of this research study. There is doubt about the represented figure 3 as the showed weigh loss in the pristine MXene before temperature 600 is weird and same behavior MEB-Air could be observed which requires repeatation of these experimental studies. In addition, the acquired/represented results should be supported with strong results. Morphology analysis as well as composition structure analysis are lacking for the composite MXene/EB which are extremely needed to check the correct synthesis of composite material. Therefore, some clear mistakes are detected throughout this experimental study which requires more validation of the reported results and should be rejected in the presented form due to lack of supporting results. Furthermore, comprehensive proof read is essential throughout the manuscript as there are a lot of typo/grammatical errors throughout the manuscript which might confuse the potential authors.

Reviewer #2 (Remarks to the Author):

The authors did well to highlight the importance that EMI shielding technologies will play in all industries, specifically highlighting the aerospace industry. The issues with MXene stability are well known, and this topic is well researched. Research in this area has focussed on increasing stability through various methods, e.g. polydopamine-MXene composites, MXene functionalisation, dispersion of MXenes in organic media. Perhaps the research into high temperature application of MXene is limited, however, the authors have not fully accounted for why MXene would be more suitable compared to existing materials available to stated applications. The focus I believe is too narrow for a communication style journal and would have benefited in applying broader research into the usage of Mxene in high temperature environment and incorporate elements such as thermal cycling, etc. Paper has merit but is not suited for Nature Communications. Below is a list of questions/suggestions concerning the manuscript:

1. Are there any SEM-EDX/STEM-EDX images to illustrate the stacking/mixed intercalating behaviour of EB and MXene?
2. Does the ratio of surface functional terminations on the MXene influence the final properties? e.g. with the synthesis method used, there may be a high number of -F terminations, whereas an in-situ MILD method might result in more -O/-OH and some -Cl terminations on top of -F terminations? - is the Ead influenced and as such is the final property of the composite influenced?

Reviewer #3 (Remarks to the Author):

The paper by Liu et al. presents a novel method to fabricate EMI shields based on Ti₃C₂T_x/exfoliated bentonite that are stable at high temperatures (even 800C). The paper is well written, the presentation of the aim of the work and of the results are well described. Results are consistent. However there are some important concerns that must be addressed prior to final acceptance of this manuscript to Nature Comms:

1. A small peak at 145 cm⁻¹ of TiO₂ can be discerned in the spectra of MEB after thermal treatment. This means that oxidation is not completely prevented in MEB. The authors shall not gloss over this aspect in the text; further comment on that should be provided. Furthermore, the

same peak is not noticed after treatment at RH90% at 400C.

2. Mechanical characterization of the membrane before and after the treatments is necessary for completeness. The provided optical images of folded specimens are not sufficient for assessing the mechanical integrity.

3. Information about the oxygen/ water barrier properties of EB should be given in order to support better the mechanism of antioxidation in MEB (e.g. experimental data or from literature).

4. In the comparison of EMI SE of MEB with other materials in the literature (Table S1 and Figure 5f), the thickness of the shield cannot be neglected since the degree of shielding depends on it. Furthermore, the authors should provide further analysis of the shielding mechanism, based on theoretical predictions of different shielding contributions (i.e. reflection, adsorption, multiple reflection). The authors are therefore suggested to read the following references on these aspects: Adv.Mater.2013, 25, 1296–1300; Nature Communications 12, 4655 (2021); Science 353, 1137-1140 (2016).

Overall, the paper can be accepted for publication after implementing the major revisions mentioned above and must be reviewed again.

Reviewer #4 (Remarks to the Author):

The authors have done an excellent job by using wide spectra of experimental and modelling evidences for validating their hypothesis on improving the thermal and ambient stability of Ti₃C₂T_x MXene through the composition with bentonite (EB) nanosheets. The data were very convincing that oxygen molecules are preferentially adsorbed on EB by composing EB sheets with MXene/EB (MEB) composite sheets. This study could open a new window for wider environmental applications of MXene composites in aqueous and high temperature conditions. Although the 2D/2D Ti₃C₂ MXene/bentonite (Bt) heterojunction was fabricate and investigated earlier for photocatalytic CO₂ reduction. <https://doi.org/10.1016/j.cej.2020.125868>, this study is still important to provide a guide to tailor the stability of MXenes to the certain environmental application.

Specifically in this study the thermally treated MEB at temperatures up to 800 °C and different RH conditions was able to retain a very good THz shielding about 50 dB at 500 °C and 48 dB at 600 °C, as compared to pristine MXene which is know to be prone to severe degradation at this conditions. I have only one "Major" concern that the authors have done the shielding measurements at room temperature after exposing the MEB to different annealing conditions. I am wondering if this is enough to make the claim on the shielding performance at high temperatures.

in order to validate that MEB can be "promising for THz shielding at high temperature in oxidizing environment for a long operation time" I see it is crucial to conduct the Terahertz shielding measurements at elevated temperatures. It is well accepted that MXenes lamellar films show different swelling/contraction profile at different temperatures. And I assume this will have a strong impact on the shielding performance at these conditions, and not necessarily comparable with the performance at room temperature.

Reviewer #5 (Remarks to the Author):

In this manuscript, Air-stable Ti₃C₂T_x based composite for high-temperature resistant terahertz shielding, has been investigated. The effect of air on the heating of Ti₃C₂ MXenes, and their further testing are investigated in detail. However, this manuscript missing important investigation related to MXene and should be further tested. Some comments are:

1. It is well known that MXenes has functional group over its surface, is there any role of MXene with bentonite to improve high-temperature resistant terahertz.

2. It has been reported that MXene was converted to TiO₂ by heating. Is there any critical role of TiO₂ to control high temperature? If it is true, then more results and such testing should be conducted. For example, presently, role of TiO₂ is critically discussed and suggested articles are Journal of Colloid and Interface Science, 2021, 591, pp. 20–37; and Applied Catalysis B: Environmental, 2021, 285, 119777

3. Author should add comparative analysis of MAX and MXene at different temperatures to see the

critical role of either morphology, structure, functional group or any other factors and some suggested article is *Journal of Alloys and Compounds*, 2020, 842, 155752

4. When MXene was mixed with Bentonite, what was the role of TiO₂, should be discussed (*Chemical Engineering Journal*, 2020, 400, 125868)

5. More critical analysis about the materials and their role should be discussed throughout the manuscript.

Response Notes

(High-temperature resistant $Ti_3C_2T_x$ based composite in air)

We would like to thank the reviewers for their constructive comments and suggestions that have helped us to make the manuscript clearer and easier to read. We have addressed all comments of the reviewers and our responses to the reviewer's comments, questions, and concerns are listed below. The changes in the revised manuscript have been highlighted in yellow.

Reviewer #1

***Comment 1.** Authors have reported an air-stable $Ti_3C_2T_x$ based composite (MEB) by compositing $Ti_3C_2T_x$ with extracted bentonite (EB) nanosheets. Based on their research study, air-stable $Ti_3C_2T_x$ MXene based composite could endure high-temperature annealing in air or humid environment. Even though, the idea behind this research study is interesting but design of experiments is done poorly and does not meet the requirements for the scientific research study. Objectives should be clear in line with adapted methodology.*

Response: We appreciate the reviewer for the efforts he/she has spent on our manuscript. Based on reviewer's comments, we have further improved our manuscript by adding the characterizations and analyses, as listed:

1. Characterization of materials

(1) Ti_3AlC_2 :

Supplementary Characterization: Morphology (SEM, XRD).

Conclusion: The Ti_3AlC_2 particle ($\approx 500 \mu m^3$) is predominately made up of square, platelet-like particles, which is consistent with previous literatures (*Adv. Mater.* 2011, 23, 4248-4253).

(2) $Ti_3C_2T_x$:

Supplementary Characterization: MILD- $Ti_3C_2T_x$ (prepared by minimally intensive layer delamination method): Morphology (SEM, XRD); the termination ratio of MILD- $Ti_3C_2T_x$ (XPS); the property of antioxidation at high-temperature (Raman).

Conclusion: In the previous manuscript, we only used HF method to synthesize $Ti_3C_2T_x$ (HF- $Ti_3C_2T_x$). Since different synthesis process may bring different ratio of terminations on $Ti_3C_2T_x$, we would like to validate if our strategy of fabricating high-temperature resistant composite is applicable on $Ti_3C_2T_x$ synthesized through other methods. To do it, we used MILD (minimally intensive layer delamination) method to synthesize $Ti_3C_2T_x$ with different termination ratios compared to $Ti_3C_2T_x$ obtained from HF method (HF- $Ti_3C_2T_x$ MXene with 38.7% =O, 31.6% -F, 7.7% -Cl, 22.0% -OH and MILD- $Ti_3C_2T_x$ with 43.8% =O, 24.4% -F, 13.1% -Cl, 18.7% -OH). As a result, MILD- $Ti_3C_2T_x$ exhibits the similar oxidation process compared with HF- $Ti_3C_2T_x$, indicating that the different termination ratios have little influence on the oxidation of $Ti_3C_2T_x$ under O_2 atmosphere at high temperature.

(3) Extracted bentonite (EB):

Supplementary Characterization: The property of antioxidation at high-temperature (TG, SEM, XRD);

Conclusion: The synthesized EB nanosheet has abundant hydroxy termination. EB displays good heat resistant property, and is stable below 700 °C. The results further confirm that EB with thermal stability plays an important role in the composite ($Ti_3C_2T_x$ MXene/EB).

(4) $Ti_3C_2T_x$ MXene/EB (MEB)

Supplementary Characterization: a. HF-MEB: morphology (STEM-EDX). b. MILD-MEB: morphology (SEM); the property of antioxidation at high-temperature (SEM, Raman).

Conclusion: According to STEM-EDX, the orientation and restacking nature of $Ti_3C_2T_x$ and EB were shown clearly. This could further support our explanation on the mechanism of high-temperature resistant property in MEB which is speculated as: i) the more compact layered structure of MEB; ii) the effect of EB on the interaction between adsorbed O_2 and $Ti_3C_2T_x$. Moreover, although the termination ratios on $Ti_3C_2T_x$ obtained by different synthesis methods were not identical, their composites with EB showed similar high-temperature resistant property, demonstrating the

universality of our strategy on suppressing the oxidation of $Ti_3C_2T_x$ at high temperature under O_2 atmosphere.

2. Applications of MEB

Supplementary Characterization: thermal stability, thermal-cycling performance.

Conclusion: the results suggest that the capabilities of fast electrothermal response and thermal-cycling stability make MEB potentially suitable for thermal/photothermal catalysis and electric/heat responsive actuator.

In summary, we have re-designed the experiment in our revised manuscript. Supplementary characterizations of samples (listed above) were conducted carefully and related analysis was added in the revised manuscript. The new results further confirm the role of EB on suppressing the oxidation process of $Ti_3C_2T_x$ at high temperature with the presence of O_2 . Furthermore, utilized as THz shielding materials, the antioxidized MEB performs high THz EMI SE of 48 dB after annealing under 600 °C in atmospheric environment, which validates MEB a promising candidate for THz shielding material that requires high-temperature working scenario. Meanwhile, we demonstrate the potential application of MEB in Joule heating and thermal-cycling, which is potentially suitable for thermal/photothermal catalysis and electric/heat responsive actuator. We are grateful for the reviewer's comments to promote us to present our work more scientifically and comprehensively. We hope our work can meet the reviewer's requirements.

Comment 2. *The adapted methodology for the combination of MXene and EB should clarify the orientation of layers clearly as the intercalation play important role for the sake of reduction in oxidation factor. Figure 1 which reflects the orientation of layers for the synthesized composite should be proven clearly with strong support as this mainly affect the purpose of this research study.*

Response: Based on the reviewer's suggestion, STEM-EDX of MEB (composition of MXene and EB) has been conducted. As shown in Fig. R1-1, layered and homogeneous distributions of Si and Ti elements indicate the stacking nature of EB and $Ti_3C_2T_x$. This is beneficial for EB to protect $Ti_3C_2T_x$ from oxidizing in the composite.

In addition, FT-IR (Fig. R1-2) and X-ray photoelectron spectroscopy (XPS, Fig. R1-3) were also used to demonstrate that EB with abundant hydroxy can interact with Ti-O termination of $\text{Ti}_3\text{C}_2\text{T}_x$ to form hydrogen bond. This would result in compact layered structure as confirmed by SEM (Fig. R1-4). We propose such a compact structure may provide a barrier to suppress the diffusion of O_2 to some extent in MEB composite, lowering the amount of O_2 around $\text{Ti}_3\text{C}_2\text{T}_x$.

Action: In the revised manuscript, we have added the STEM-EDX images of MEB in Fig. 2 at page 5, the SEM images of $\text{Ti}_3\text{C}_2\text{T}_x$ plate and EB plate in Fig. S2. Besides, relevant analysis is added to the revised manuscript at page 4:

“The successful synthesis of those two materials was confirmed by X-ray diffraction (XRD) patterns (Fig. S1), scanning electron microscopy (SEM) and transmission electron microscopy (TEM) images (Fig. S2), which show typical nanosheet morphology with lateral size of $\sim 8 \mu\text{m}$ for $\text{Ti}_3\text{C}_2\text{T}_x$ and $2 \sim 5 \mu\text{m}$ for EB, respectively.”

“MEB maintains the similar lamellar structure and flexibility compared to the restacked $\text{Ti}_3\text{C}_2\text{T}_x$ and EB films (Fig. S4), which is further confirmed with the images of scanning transmission electron microscopy (STEM). As shown in Fig. 2a, layered and homogeneous distributions of Si and Ti elements indicate the stacking of EB and $\text{Ti}_3\text{C}_2\text{T}_x$.”

Fig. R1-1 Cross-sectional STEM image of MEB with EDX mapping (**Fig. 2a** in revised manuscript).

Fig. R1-2 Fourier transform infrared spectroscopy (FT-IR) of $\text{Ti}_3\text{C}_2\text{T}_x$, EB, MEB films. The strength of ν (OH) and γ (OH) peaks on MEB are higher than pure $\text{Ti}_3\text{C}_2\text{T}_x$, due to the composing with EB which has abundant hydroxy. (**Fig. S20** in revised manuscript)

Fig. R1-3 X-ray photoelectron spectroscopy (XPS) of O 1s of $\text{Ti}_3\text{C}_2\text{T}_x$ -RT and MEB-RT. (**Fig. 3e** in revised manuscript)

Fig. R1-4 Photograph of the MEB film exhibiting the flexibility and the corresponding cross-sectional SEM image (**Fig. S4b** in revised manuscript).

Comment 3. *There is doubt about the represented figure 3 as the showed weigh loss in the pristine MXene before temperature 600 is weird and same behavior MEB-Air could be observed which requires repeation of these experimental studies. In addition, the acquired/represented results should be supported with strong results.*

Response: According to the reviewer's comments, we have made an explanation:

As shown in Fig. R1-5a, $\text{Ti}_3\text{C}_2\text{T}_x$ exhibits four stages of weight change under synthetic air (with volume fractions of 21% O_2 and 79% N_2) at 100 °C, 360 °C, 480 °C and 600 °C, which is consistent with the previous reports (*ACS Appl. Nano Mater.* 2020, 3, 3195-3204). In detail, at 100 °C, the removal of intercalated water leads to the appearance of first H_2O peak and a weight loss. The second weak H_2O peak is observed at 360 °C due to the slow dissociation of hydroxyl surface terminations. Around 480 °C, the weight of $\text{Ti}_3\text{C}_2\text{T}_x$ quickly increases due to the interaction with O_2 , which evolves into TiO_2 accompanied by the generation of CO_2 . The oxidation process accelerates with the increase of annealing temperature and approaches the completion at 600 °C. MEB annealed in air (Fig. R1-5b), exhibits different behavior in TG-MS: i) the weight of MEB starts to increase at 650 °C with the appearance of CO_2 , which is much higher than pure $\text{Ti}_3\text{C}_2\text{T}_x$ of 480 °C; ii) $\text{Ti}_3\text{C}_2\text{T}_x$ in MEB transforms into TiO_2 completely at 920 °C with the strongest signal of CO_2 , while 600 °C is enough for $\text{Ti}_3\text{C}_2\text{T}_x$. In addition, the TG-MS analysis of $\text{Ti}_3\text{C}_2\text{T}_x$ in Ar is adapted, as shown in Fig. R1-6. The three stages

of weight loss at 100 °C, 300 °C and 900 °C are the release of interlayered H₂O, -OH and =O respectively, which is consistent with Gogotsi's report (*Chem. Mater.* 2019, 31, 3324-3332; *J. Mater. Chem. A* 2020, 8, 573-581). Besides, the TG-MS results were conducted by randomly choosing samples without “exquisitely” selection, we believe there is no doubt about the repetition. Therefore, the Ti₃C₂T_x we synthesized is normal and the TG-MS analysis process is logical. We reasonably speculate that the induction of EB does affect the interaction between adsorbed O₂ and Ti₃C₂T_x.

Fig. R1-5 Thermal gravimetric (TG) curves in air with mass spectrometry analysis (MS) for the atomic mass unit (amu) of 18/H₂O and 44/CO₂ for (a) Ti₃C₂T_x and (b) MEB. (**Fig. 3a, c** in revised manuscript).

Fig. R1-6 TG curves of Ti₃C₂T_x in Ar with mass spectrometry analysis (MS) for the atomic mass unit (amu) of 18/H₂O and 44/CO₂. (**Fig. S10a** in revised manuscript).

Comment 4. Morphology analysis as well as composition structure analysis are lacking for the composite MXene/EB which are extremely needed to check the correct synthesis of composite material. Therefore, some clear mistakes are detected throughout this experimental study which requires more validation of the reported results and should be rejected in the presented form due to lack of supporting results.

Response: We totally agree that the morphology and composition structure analysis are extremely important for this work. Following the reviewer’s suggestion, we’ve added several critical characterizations and revised the related analysis. For clear presentation, the characterization techniques, samples and related Figures in revised manuscript and Supporting Information in our work are summarized in Table R1-1:

Table R1-1 The critical analysis about the materials in our work

Material	Analysis (Fig.)								
	SEM	TEM/STEM-EDX	XRD	XPS	Raman	TG	FT-IR	Mechanical Integrity	Zeta potential
Ti₃AlC₂	Fig.S31		Fig.S32						
	Fig.2								
	Fig.S2								
	Fig.S4		Fig.S1	Fig.3					
Ti₃C₂T_x	Fig.S8	Fig.2	Fig.S9	Fig.4	Fig.2	Fig.3		Fig.2	
	Fig.S23	Fig.S2	Fig.S24	Fig.S19	Fig.4	Fig.S10	Fig.S20	Fig.S6	Fig.S3
	Fig.S24		Fig.S25	Fig.S26	Fig.S21				
	Fig.S25								
	Fig.S27								
EB	Fig.S2		Fig.S1			Fig.3			
	Fig.S4		Fig.S11			Fig.S10	Fig.S20		Fig.S3
	Fig.S12	Fig.S2							
MEB	Fig.2							Fig.2	
	Fig.S4								
	Fig.S7	Fig.2	Fig.S9	Fig.3	Fig.2	Fig.3		Fig.S5	Fig.S3
	Fig.S22		Fig.S24	Fig.S19	Fig.4		Fig.S20	Fig.S6	
	Fig.S24				Fig.S21				
	Fig.S27								
h-BN	Fig.S15		Fig.S15		Fig.S18				
MBN	Fig.S17				Fig.S18				

The characterizations of materials we added in our revised manuscript are marked in red in Table R1-1.

In conclusion, we prepared $\text{Ti}_3\text{C}_2\text{T}_x$ based composite and did sufficient characterizations of each material in our revised manuscript (as shown in Table R1-1). The above analyses further confirm the suppression on oxidation process of $\text{Ti}_3\text{C}_2\text{T}_x$ at high temperature caused by the introduction of EB in $\text{Ti}_3\text{C}_2\text{T}_x$ composite.

Comment 5. Furthermore, comprehensive proof read is essential throughout the manuscript as there are a lot of typo/grammatical errors throughout the manuscript which might confuse the potential authors.

Response: We thank the reviewer for pointing this out. We have gone through the manuscript and corrected the typo/grammatical errors.

Reviewer #2

Comment 1. *The authors did well to highlight the importance that EMI shielding technologies will play in all industries, specifically highlighting the aerospace industry. The issues with MXene stability are well known, and this topic is well researched. Research in this area has focussed on increasing stability through various methods, e.g. polydopamine-MXene composites, MXene functionalisation, dispersion of MXenes in organic media. Perhaps the research into high temperature application of MXene is limited, however, the authors have not fully accounted for why MXene would be more suitable compared to existing materials available to stated applications.*

Response: Thanks for the reviewer's critical comments. The reason why MXene would be more suitable compared to existing materials available for EMI shielding is indeed insufficiently displayed in our previous manuscript. Following the reviewer's suggestion, we have made an explanation:

Metallic conductivity and good mechanical properties coupled with hydrophilicity make MXenes a good candidate as EMI shielding material, especially in microwave band (*Science* 2016, 353, 1137-1140). To satisfy the common commercial EMI shielding requirements (above 20 dB), large thicknesses, usually more than 1 mm, are needed for carbon-based and metal-based materials (such as rGO/PS, GO/PU, Ni foam, rGO/Fe₃O₄ etc.) (*Science* 2016, 353, 1137-1140; *Adv. Funct. Mater.* 2015, 25, 559-566). Although graphene films exhibited enhanced EMI SE performance with a thinner thickness of 50 μm (*Carbon* 2015, 82, 353-359) compared to conventional dense metallic films or foils, the hydrophobic surface, and complex preparing process may be the barriers for graphene to design as THz EMI materials (*Adv. Mater.* 2013, 25, 1296-1300; *Nat. Commun.* 2021, 12, 4655; *ACS Nano* 2020, 14, 8754-8765; *ACS Nano* 2021, 15, 829-839). The EMI shielding performances of various shielding materials are displayed in Table R2-1, in which most carbon-based materials and metals show high SE with large thickness (0.35-60 mm). Nevertheless, the ultrathin MXene film (a 24-layer film of ≈ 55 nm thickness) reported by Gogotsi et al. demonstrates 99% reflection shielding (EMI SE of 20 dB) (*Adv. Mater.* 2020, 32) and a Ti₃C₂T_x film of 45 μm shows ultrahigh SE of 92 dB (*Science* 2016, 353, 1137-1140). Based on the hydrophilicity of

MXene, Han *et al.* composited $Ti_3C_2T_x$ with wax, which delivers an EMI SE value of 76.1 dB with a thickness of 1 mm (*ACS Appl. Mater. Interfaces* 2016, 8, 21011-21019). Therefore, MXenes possess an excellent capability of being EMI shielding materials which makes it reasonable for our choice.

Table R2-1 EMI shielding performance of various shielding materials

EMI Shielding Materials	Filler (wt.%)	Matrix	Thickness (mm)	EMI SE (dB)	Ref.
rGO	7	PS	2.5	45.1	1
rGO	10	PEI	2.3	22	2
rGO/MG	60 (Vol. %)	PVA	0.35	27	3
GO/MWCNTs	1	PDMS	15	32	4
GO	10	PU	60	39.4	5
GO	5	PMMA	2.4	19	6
Graphene paper	Bulk		0.05	60	7
rGO/Fe ₃ O ₄	10	PVC	1.8	13	8
rGO/MnO ₂	Bulk		3	57	9
CNT/GO	0.78	PDMS	1.6	38.4	10
rGO-BaTiO ₃	Bulk		1.5	41.7	11
CCA@ rGO	3.05	PDMS	15	51	12
CF	10(Vol. %)	PP	3.1	24.9	13
Cu _x S	45.23	PAN	0.423	29	14
Ag Nanowires	2.5(Vol. %)	PS	0.8	33	15
MoS ₂	50		3.5	45.5	16
Ni	40(Vol. %)	PVDF	1.95	23	17
Stainless-Steel	1.1(Vol. %)	PP	3.1	48	18
Ti ₃ C ₂ T _x	50	wax	1	76.1	19
Mo ₂ TiC ₂ T _x	Bulk		0.004	23	20
Ti ₃ C ₂ T _x	Bulk		0.045	92	20
Ti ₃ C ₂ T _x	Bulk		0.055	20	21

References in Table R2-1

1. Yan, D.-X. *et al.* Structured reduced graphene oxide/polymer composites for ultra-efficient electromagnetic interference shielding. *Adv. Funct. Mater.* **25**, 559-566 (2015).
2. Ling, J. *et al.* Facile preparation of lightweight microcellular polyetherimide/graphene composite foams for electromagnetic interference shielding. *ACS Appl. Mater. Interfaces* **5**, 2677-2684 (2013).
3. Yuan, B. *et al.* Design of artificial nacre-like hybrid films as shielding to mitigate electromagnetic pollution. *Carbon* **75**, 178-189 (2014).
4. Jia, H. *et al.* 3D graphene/carbon nanotubes/polydimethylsiloxane composites as high-performance electromagnetic shielding material in X-band. *Compos. Part A Appl. Sci. Manuf.* **129**, 105712 (2020).
5. Shen, B., Li, Y., Zhai, W. & Zheng, W. Compressible graphene-coated polymer foams with ultralow density for adjustable electromagnetic interference (EMI) shielding. *ACS Appl. Mater. Interfaces* **8**, 8050-8057 (2016).
6. Zhang, H. B., Yan, Q., Zheng, W. G., He, Z. & Yu, Z. Z. Tough graphene-polymer microcellular foams for electromagnetic interference shielding. *ACS Appl. Mater. Interfaces* **3**, 918-924 (2011).
7. Zhang, L. *et al.* Preparation and characterization of graphene paper for electromagnetic interference shielding. *Carbon* **82**, 353-359 (2015).
8. Yao, K. *et al.* Flammability properties and electromagnetic interference shielding of PVC/graphene composites containing Fe₃O₄ nanoparticles. *RSC Advances* **5**, 31910-31919 (2015).
9. Gupta, T. K. *et al.* MnO₂ decorated graphene nanoribbons with superior permittivity and excellent microwave shielding properties. *J. Mater. Chem. A* **2**, 4256 (2014).
10. Song, Q. *et al.* Carbon nanotube-multilayered graphene edge plane core-shell hybrid foams for ultrahigh-performance electromagnetic-interference shielding. *Adv. Mater.* **29**, 1701583 (2017).
11. Yuchang, Q., Qinlong, W., Fa, L., Wancheng, Z. & Dongmei, Z. Graphene nanosheets/BaTiO₃ ceramics as highly efficient electromagnetic interference shielding materials in the X-band. *J. Mater. Chem. C* **4**, 371-375 (2016).
12. Song, P. *et al.* Lightweight, flexible cellulose-derived carbon aerogel@reduced graphene oxide/PDMS composites with outstanding EMI shielding performances and excellent thermal conductivities. *Nanomicro Lett.* **13**, 91 (2021).
13. Ameli, A., Jung, P. U. & Park, C. B. Electrical properties and electromagnetic interference shielding effectiveness of polypropylene/carbon fiber composite foams. *Carbon* **60**, 379-

391 (2013).

14. Li, H. *et al.* CuxS/PAN 3D nanofiber mats as ultra-lightweight and flexible electromagnetic interference shielding materials. *Macromol. Mater. Eng.* **304**, 1900482 (2019).
15. Arjmand, M., Moud, A. A., Li, Y. & Sundararaj, U. Outstanding electromagnetic interference shielding of silver nanowires: comparison with carbon nanotubes. *RSC Advances* **5**, 56590-56598 (2015).
16. Ning, M. *et al.* Phase manipulating toward molybdenum disulfide for optimizing electromagnetic wave absorbing in gigahertz. *Adv. Funct. Mater.* **31**, 2011229 (2021).
17. Gargama, H., Thakur, A. K. & Chaturvedi, S. K. Polyvinylidene fluoride/nickel composite materials for charge storing, electromagnetic interference absorption, and shielding applications. *J. Appl. Phys.* **117**, 224903 (2015).
18. Ameli, A., Nofar, M., Wang, S. & Park, C. B. Lightweight polypropylene/stainless-steel fiber composite foams with low percolation for efficient electromagnetic interference shielding. *ACS Appl. Mater. Interfaces* **6**, 11091-11100 (2014).
19. Han, M. *et al.* Ti₃C₂ MXenes with modified surface for high-performance electromagnetic absorption and shielding in the X-Band. *ACS Appl. Mater. Interfaces* **8**, 21011-21019 (2016).
20. Shahzad, F. *et al.* Electromagnetic interference shielding with 2D transition metal carbides (MXene). *Science* **353**, 1137-1140 (2016).
21. Yun, T. *et al.* Electromagnetic shielding of monolayer mxene assemblies. *Adv. Mater.* **32**, 1906769 (2020).

Action: We have added the description in the Introduction part of the revised manuscript, page 2:

“For instance, compared to the conventional dense metallic films, and graphene with hydrophobic surface^{4,5}, self-assembled ultrathin MXene film (a 24-layer film of ≈ 55 nm thickness) is capable of providing 99% reflection shielding (EMI SE of 20 dB) in microwave band⁶. Notably, besides the traditional microwave band, MXenes are demonstrated as high efficient shielding or absorbing material for higher-frequency terahertz wave (THz, frequency ranging in 0.1-10 THz) which is assigned for 6G wireless communication and has many potential applications in aerospace industry⁷⁻¹⁰.”

Reference:

4. Pavlou, C. *et al.* Effective EMI shielding behaviour of thin graphene/PMMA nanolaminates in the THz range. *Nat. Commun.* **12**, 4655 (2021).

5. Chen, Z., Xu, C., Ma, C., Ren, W. & Cheng, H. M. Lightweight and flexible graphene foam composites for high-performance electromagnetic interference shielding. *Adv. Mater.* **25**, 1296-1300 (2013).
6. Yun, T. *et al.* Electromagnetic shielding of monolayer MXene assemblies. *Adv. Mater.* **32**, 1906769 (2020).
7. Shahzad, F. *et al.* Electromagnetic interference shielding with 2D transition metal carbides (MXene). *Science* **353**, 1137-1140 (2016).
8. Li, X. *et al.* 3D Seed-germination-like MXene with in situ growing CNTs/Ni heterojunction for enhanced microwave absorption via polarization and magnetization. *Nano-Micro Lett.* **13**, 157 (2021).
9. Shui, W. *et al.* Ti₃C₂T_x MXene sponge composite as broadband terahertz absorber. *Adv. Opt. Mater.* **8**, 2001120 (2020).
10. Wan, H., Liu, N., Tang, J., Wen, Q. & Xiao, X. Substrate-independent Ti₃C₂T_x MXene waterborne paint for terahertz absorption and shielding. *ACS Nano* **15**, 13646–13652 (2021).

Comment 2. *The focus I believe is too narrow for a communication style journal and would have benefited in applying broader research into the usage of Mxene in high temperature environment and incorporate elements such as thermal cycling, etc. Paper has merit but is not suited for Nature Communications.*

Response: We thank the reviewer for this constructive comment. Potential applications of MEB film besides THz EMI shielding were added in the revised manuscript. First, the prepared MEB films with high conductivity (850 S cm⁻¹) also display excellent Joule heating performance. As is shown in Fig. R2-1, the heating rate of MEB film can be 20 °C s⁻¹ under a driving voltage of 3.0 V, with a steady-state temperature of 198 °C without large temperature fluctuation for more than 3.5 hours, which reflects the rapid thermal response and stable high-temperature electrothermal performance of MEB. As a contrast, pristine Ti₃C₂T_x film reaches to 190 °C at the same voltage and is stable for only 0.57 h, then the temperature begins to decline due to the rapid oxidation of Ti₃C₂T_x.

In addition, MEB exhibits a stable thermal cycling performance as shown in Fig. R2-2. We recorded the steady-state temperatures of MEB under the applied voltage of 3V and after cut-off of voltage. This process of voltage being on/off is repeated on MEB

for 30 times during 4500 s. The steady-state temperatures of MEB under voltage being on/off is basically the same during cycling, which indicates stable thermal-cycling of MEB.

Therefore, the fast electrothermal response and thermal-cycling stability at high temperature offer the possibility of using MEB as thermal/photothermal catalyst and electric/heat responsive actuator.

Action: We have added the thermal cycling and the Joule heating performances of MEB in the revised manuscript, page 13:

“To further explore the potential application of MEB by virtue of the high-temperature resistant property, we also measure the Joule heating performance of MEB. Joule heating device is required to provide fast heating and high temperature output (in some cases, hundreds of centigrade degree are needed). In general, the metallic conductivity of $Ti_3C_2T_x$ meets the demand for Joule heating device. As shown in Fig. 5d, pristine $Ti_3C_2T_x$ film quickly reaches to 190 °C under a driving voltage of 3.0 V. However, the temperature can only be kept for 0.57 h, then begins to decline due to the oxidation of $Ti_3C_2T_x$. As a comparison, the heating rate of MEB film can be 20 °C s⁻¹, with a steady-state temperature of 198 °C without large temperature fluctuation for more than 3.5 hours, which reflects the rapid thermal response and stable high-temperature electrothermal performance. In addition, the thermal cycling performance of MEB is tested as shown in Fig. 5e. The steady-state temperatures of MEB under driving voltage being on/off is basically identical during 30 cycles, which reveals the stable thermal-cycling of MEB. The above results suggest that the capabilities of fast thermal response and thermal-cycling stability make MEB potentially suitable for thermal/photothermal catalysis and beyond.”

Fig. R2-1 Long-term Joule heating performance of MEB and $\text{Ti}_3\text{C}_2\text{T}_x$ films driven by an applied voltage of 3.0 V. (**Fig. 5d** in revised manuscript)

Fig. R2-2 The thermal cycling performance of MEB film, which is realized by switching the applied voltage (3.0 V) on and off repeatedly. (**Fig. 5e** in revised manuscript)

Comment 3. Are there any SEM-EDX/STEM-EDX images to illustrate the stacking/mixed intercalating behaviour of EB and MXene?

Response: We thank the reviewer for this constructive suggestion. Based on the reviewer’s suggestion, STEM-EDX of MEB (composition of $\text{Ti}_3\text{C}_2\text{T}_x$ MXene and EB) has been conducted. As shown in Fig. R2-3, layered and homogeneous distributions of Si and Ti elements indicate the stacking of EB and $\text{Ti}_3\text{C}_2\text{T}_x$ in the MEB composite.

Action: We have added the STEM-EDX images of MEB in Fig. 2 at page 5, the SEM images of $\text{Ti}_3\text{C}_2\text{T}_x$ plate and EB plate in Fig. S2, and added relevant analysis in the revised manuscript at page 4:

“The successful synthesis of those two materials was confirmed by X-ray diffraction

(XRD) patterns (Fig. S1), scanning electron microscopy (SEM) and transmission electron microscopy (TEM) images (Fig. S2), which show typical nanosheet morphology with lateral size of $\sim 8 \mu\text{m}$ for $\text{Ti}_3\text{C}_2\text{T}_x$ and $2 \sim 5 \mu\text{m}$ for EB, respectively.”

“MEB maintains the similar lamellar structure and flexibility compared to the restacked $\text{Ti}_3\text{C}_2\text{T}_x$ and EB films (Fig. S4), which is further confirmed with the images of scanning transmission electron microscopy (STEM). As shown in Fig. 2a, layered and homogeneous distributions of Si and Ti elements indicate the stacking of EB and $\text{Ti}_3\text{C}_2\text{T}_x$.”

Fig. R2-3 Cross-sectional STEM images of MEB with EDX mapping (**Fig. 2a** in revised manuscript).

Comment 4. *Does the ratio of surface functional terminations on the MXene influence the final properties? e.g. with the synthesis method used, there may be a high number of -F terminations, whereas an in-situ MILD method might result in more -O/-OH and some -Cl terminations on top of -F terminations? - is the E_{ad} influenced and as such is the final property of the composite influenced?*

Response: We thank the reviewer for this constructive suggestion. In general, the surface termination affects the chemical and physical properties of MXene. Therefore, to validate the influence of termination on the final high-temperature resistant property, we prepared $\text{Ti}_3\text{C}_2\text{T}_x$ through two approaches which shows different termination ratios: **HF- $\text{Ti}_3\text{C}_2\text{T}_x$:** $\text{Ti}_3\text{C}_2\text{T}_x$ was synthesized by selectively etching Al of Ti_3AlC_2 MAX using

HF and HCl (*Adv. Energy Mater.* 2020, 11; *ACS Nano* 2021,15, 13646–13652). 1 g of Ti_3AlC_2 powder (11 Technology Co.) was added to the etchant solution, which includes 12 mL 38 wt.% hydrochloric acid (HCl, Sigma-Aldrich), 2 mL 49 wt.% hydrofluoric acid (HF, Sigma-Aldrich) and 6 mL deionized (DI) water. Next, the Ti_3AlC_2 powder was etched for 24 h by string at 400 rpm at 35 °C. Then, the multilayered $\text{Ti}_3\text{C}_2\text{T}_x$ was obtained by washing the reaction mixture with DI water via centrifugation at 3500 rpm (5 min per cycle). After washing for 5 ~ 6 times (until the $\text{pH} \geq 6$), the sediment was dispersed into the delamination solution, which was prepared by adding 1 g of LiCl (Sigma-Aldrich) into 40 mL DI water. Then, the mixture was stirred for 4 h at room temperature. The dispersion was washed using DI water and the single layer $\text{Ti}_3\text{C}_2\text{T}_x$ aqueous solution was obtained by centrifugation at 3500 rpm for 5 min.

MILD- $\text{Ti}_3\text{C}_2\text{T}_x$: LiF (1.6 g; Sigma-Aldrich) was added to 9 M HCl (20 ml; HCl, 38wt.%, Sigma-Aldrich) and continuously stirred at 400 rpm for 30 minutes at 40 °C. 1g Ti_3AlC_2 (11 Technology Co.) was added into the mixture solution and stirred at 400 rpm for 24 h at 40 °C. Then the dispersion was washed by DI water for 4 ~ 6 times until the $\text{pH} \geq 6$. After washing, the sediment was dispersed in 50 mL fresh DI water and the single layer $\text{Ti}_3\text{C}_2\text{T}_x$ aqueous solution was obtained by centrifugation at 3500 rpm for 5 min (*Adv. Mater.* 2020, 32, e2001093).

The termination analysis of HF- and MILD- $\text{Ti}_3\text{C}_2\text{T}_x$: The termination analysis of HF- $\text{Ti}_3\text{C}_2\text{T}_x$ and MILD- $\text{Ti}_3\text{C}_2\text{T}_x$ (characterized by SEM and XRD, Fig. R2-4) was conducted through XPS as shown in Fig. R2-5. By fitting the relative intensities of the O 1s, F 1s, Cl 2p peak (*ACS Nano* 2019, 13, 9449-9456; *Chem. Mater.* 2019, 31, 6590-6597), we determine the termination of HF- $\text{Ti}_3\text{C}_2\text{T}_x$ MXene (T_x) to be 38.7% =O, 31.6% -F, 7.7% -Cl, 22.0% -OH and MILD- $\text{Ti}_3\text{C}_2\text{T}_x$ with 43.8% =O, 24.4% -F, 13.1% -Cl, 18.7% -OH, which reveals the different termination ratios for two samples.

Characterization of pristine MILD $\text{Ti}_3\text{C}_2\text{T}_x$ and MILD MEB (MILD $\text{Ti}_3\text{C}_2\text{T}_x$ /EB composite) after high-temperature annealing in oxygen atmosphere: We investigated the antioxidation of MILD $\text{Ti}_3\text{C}_2\text{T}_x$ and MILD MEB by the same treatment with $\text{Ti}_3\text{C}_2\text{T}_x$ and MEB (displayed in our original manuscript). Similarly, the samples

were annealed at 400°C under synthetic air (with volume fractions of 21% O₂ and 79% N₂) for different time, which are denoted as MILD Ti₃C₂T_x/MEB-Air-400C-t (t is annealing time, hours). The cross-sectional SEM images (Fig. R2-6) show a good retention of layered structure for MILD MEB-Air-400C-2, 4, 6. At the same time, TiO₂ particle is obvious in MILD Ti₃C₂T_x-Air-400C-2, 4, 6, and the degrading process is accelerated by elevated temperature. The oxidation of different samples is further confirmed by Raman spectra displayed in Fig. R2-7. Typical E_g (Ti, C) mode at 289, 369 cm⁻¹ and A_{1g} (Ti, C) mode at 201, 599, 722 cm⁻¹ of Ti₃C₂T_x remain high strength in MILD MEB-Air-400C-2 and A_{1g} peak (201 cm⁻¹) is still obvious in MILD MEB-Air-400C-6. By contrast, MILD Ti₃C₂T_x-Air-400C-2 shows the transformation of Ti-C bonding to Ti-O bonding, which is indicated by the appearance of TiO₂ peaks at 145, 400, 519, 637 cm⁻¹. It confirms the suppression of oxidation for MILD MEB at high temperature in air. In summary, MEB and MILD MEB with different termination ratios on Ti₃C₂T_x display similar high-temperature resistant property in oxygen atmosphere. It reveals that the terminations on Ti₃C₂T_x may have little influence on the high-temperature resistant property of the MEB composite.

Theoretical calculations: To further investigate the terminational influence and the mechanism of high-temperature resistant property in MEB, theoretical calculations were performed based on density functional theory (DFT). The adsorption energy (E_{ad}) and Bader charge states of O₂ molecule adsorbed on Ti₃C₂O₂ and EB substrates were analyzed in the original manuscript. The F-termination is another type of surface saturation on bare MXene, accounting for a high proportion of T_x. Therefore, the calculated E_{ad} and Bader charge states of Ti₃C₂F₂ before and after adsorption of O₂ were supplemented in revision.

The E_{ad} of O₂ molecule adsorbed on Ti₃C₂F₂ is -0.030 eV, less negative than that on EB (-0.916 eV). This is in line with the transferred charges: the adsorbed O₂ obtains 0.421 electron from EB, much larger than that on Ti₃C₂F₂ (0.052 electron) (Fig. R2-8), further confirming the stronger binding ability between O₂ and EB. The results reveal that Ti₃C₂F₂ exhibits more inferior interaction with O₂ to EB, suggestive of similar behavior to Ti₃C₂O₂.

Action: (a) the calculated E_{ad} and Bader charge states of $Ti_3C_2F_2$ before and after adsorption of O_2 were supplemented in our revise manuscript, page 8:

“The F-terminated MXene is also calculated as F termination accounts for a high proportion of T_x . The calculated E_{ad} and Bader charge states of $Ti_3C_2F_2$ before and after adsorption of O_2 show that $Ti_3C_2F_2$ exhibits more inferior interaction with O_2 compared to EB (Fig. S14), suggestive of similar behavior to that of $Ti_3C_2O_2$.”

(b) We have added the synthesis method of MILD MEB and relevant analysis in the revised manuscript, page 10-11. In addition, we added the related figures and the analysis to the revised Supplementary Information.

“The effect of termination ratio on high-temperature resistant property of MEB.

To detect the influence of terminations on the high-temperature resistant property of $Ti_3C_2T_x$, two approaches are adopted to synthesize $Ti_3C_2T_x$, which are minimally intensive layer delamination method (denoted to MILD- $Ti_3C_2T_x$, Fig. S25) and HF etching method (HF- $Ti_3C_2T_x$), respectively. By fitting the relative intensities of the O 1s, F 1s, Cl 2p peaks^{36,37} as shown in Fig. 4a-d and S26, we determine the termination of HF- $Ti_3C_2T_x$ to be 38.7% =O, 31.6% -F, 7.7% -Cl, 22.0% -OH and MILD- $Ti_3C_2T_x$ with 43.8% =O, 24.4% -F, 13.1% -Cl, 18.7% -OH, which reveals the different termination ratios for two samples. For clear description, composite consisting of MILD- $Ti_3C_2T_x$ and EB is denoted as MILD MEB in order to differentiate with MEB (HF- $Ti_3C_2T_x$ /EB). Following the same thermal treatment of MEB, MILD MEB-Air-400C-2 shows high-intensive E_g (Ti, C) peaks at 289, 369 cm^{-1} and A_{1g} (Ti, C) peaks at 201, 599, 722 cm^{-1} of $Ti_3C_2T_x$, and A_{1g} peak (201 cm^{-1}) is still obvious in MILD MEB-Air-400C-6 as shown in Fig. 4f. By contrast, MILD $Ti_3C_2T_x$ -Air-400C-2 (Fig. 4e) shows the transformation of Ti-C bonding to Ti-O bonding, which is indicated by the appearance of TiO_2 peaks at 145, 400, 519, 637 cm^{-1} . Along with the SEM analysis (Fig. S27), the results confirm the high-temperature resistant property for MILD MEB in air. The similar behavior in MEB and MILD MEB reveals our strategy on suppressing the oxidation of $Ti_3C_2T_x$ is independent on the termination ratio of $Ti_3C_2T_x$.”

“MILD- $Ti_3C_2T_x$ (prepared by minimally intensive layer delamination method): LiF (1.6 g; Sigma-Aldrich) was added to 9 M HCl (20 ml; HCl, 38wt.%, Sigma-Aldrich) and

continuously stirred at 400 rpm for 30 minutes at 40 °C. 1g Ti_3AlC_2 (11 Technology Co.) was added into the mixture solution and stirred at 400 rpm for 24 h at 40 °C. Then the dispersion was washed by DI water for 4 ~ 6 times until the $\text{pH} \geq 6$. After washing, the sediment was dispersed in 50 mL fresh DI water and the single layer $\text{Ti}_3\text{C}_2\text{T}_x$ aqueous solution was obtained by centrifugation at 3500 rpm for 5 min⁴⁰.”

“Meanwhile, MILD $\text{Ti}_3\text{C}_2\text{T}_x$ film and MILD MEB are fabricated by MILD $\text{Ti}_3\text{C}_2\text{T}_x$ solution in the similar process.”

Reference:

36. Chen, X. *et al.* N-Butyllithium-treated $\text{Ti}_3\text{C}_2\text{T}_x$ MXene with excellent pseudocapacitor performance. *ACS Nano* **13**, 9449-9456 (2019).
37. Schultz, T. *et al.* Surface termination dependent work function and electronic properties of $\text{Ti}_3\text{C}_2\text{T}_x$ MXene. *Chem. Mater.* **31**, 6590-6597 (2019).
40. Zhang, J. *et al.* Scalable manufacturing of free-standing, strong $\text{Ti}_3\text{C}_2\text{T}_x$ MXene films with outstanding conductivity. *Adv. Mater.* **32**, e2001093 (2020).

Fig. R2-4. (a) SEM image of MILD- $\text{Ti}_3\text{C}_2\text{T}_x$ flake; (b) XRD pattern of MILD- $\text{Ti}_3\text{C}_2\text{T}_x$. The results show typical nanosheet morphology with lateral size of $\sim 12 \mu\text{m}$ for $\text{Ti}_3\text{C}_2\text{T}_x$. (**Fig. S25** in revised manuscript)

Fig. R2-5. (a-c) XPS of O 1s, F 1s and Cl 2p signals, respectively, for the pristine HF-Ti₃C₂T_x and MILD-Ti₃C₂T_x films. (d) XPS survey spectra of HF-Ti₃C₂T_x and MILD-Ti₃C₂T_x films with the ratio of O, F, Cl as a elements list (inserted). (e) The termination ratio of HF-Ti₃C₂T_x and MILD-Ti₃C₂T_x measured by XPS with a table list (inserted). (**Fig. 4a-d and Fig. S26** in revised manuscript)

Fig. R2-6. SEM images of the cross-section of different samples: (a1) MILD $\text{Ti}_3\text{C}_2\text{T}_x$ -Air-400C-2, (b1) MILD $\text{Ti}_3\text{C}_2\text{T}_x$ -Air-400C-4, (c1) MILD $\text{Ti}_3\text{C}_2\text{T}_x$ -Air-400C-6; (a2) MILD MEB-Air-400C-2, (b2) MILD MEB-Air-400C-4, (c2) MILD MEB-Air-400C-6. (Fig. S27 in revised manuscript)

Fig. R2-7. (a) Raman spectra of MILD $\text{Ti}_3\text{C}_2\text{T}_x$ before and after treatment under synthetic air at 400 °C for 2, 4, 6 hours. (b) Raman spectra of MILD MEB before and after treatment under synthetic air at 400 °C for 2, 4, 6 hours. (Fig. 4e-f in revised manuscript)

Fig. R2-8 The charge density difference plots for O₂ adsorbed on Ti₃C₂F₂: (a) side view, (b) top view. The isosurface level is set to be 0.0002 e/Å³. (**Fig. S14** in revised manuscript)

Reviewer #3

Comment 1. *The paper by Liu et al. presents a novel method to fabricate EMI shields based on Ti₃C₂T_x/exfoliated bentonite that are stable at high temperatures (even 800C). The paper is well written, the presentation of the aim of the work and of the results are well described. Results are consistent. However there are some important concerns that must be addressed prior to final acceptance of this manuscript to Nature Comms:*

A small peak at 145 cm⁻¹ of TiO₂ can be discerned in the spectra of MEB after thermal treatment. This means that oxidation is not completely prevented in MEB. The authors shall not gloss over this aspect in the text; further comment on that should be provided. Furthermore, the same peak is not noticed after treatment at RH90% at 400C.

Response: We appreciate the efforts the reviewer has spent on our manuscript and thanks for the reviewer's positive comments. In our work, we report air-stable Ti₃C₂T_x MXene/EB (MEB) that could endure high-temperature annealing in air or humid environment. The introduce of EB can suppress the oxidation process of Ti₃C₂T_x, but cannot completely prevent it in MEB. The concern the reviewer raised may be caused by our ambiguous expression of "antioxidation" in our manuscript. Therefore, we have made some modifications in the revised manuscript. According to the reviewer's suggestions, we also made an explanation about the comments:

As shown in Raman spectra (Fig. R3-1), E_g (Ti, C) mode at 289, 369 cm⁻¹ and A_{1g} (Ti, C) mode at 201, 599, 722 cm⁻¹ are the typical peaks of Ti₃C₂T_x. The signals of TiO₂ appear at 145, 400, 519, 637 cm⁻¹ are obvious in Ti₃C₂T_x-Air-400C-2. By contrast, MEB-Air-400C-2 shows high-intensity A_{1g} (Ti, C) peaks at 201 and 599 cm⁻¹, as well as relatively weaker E_g (Ti, O) at 145 cm⁻¹ compared to Ti₃C₂T_x-Air-400C-2. It reveals that EB plays an important role in suppressing the oxidizing process of Ti₃C₂T_x, which is also proved by the Raman spectrum of MEB-Air-400C-6 with Ti₃C₂T_x signals at 145 cm⁻¹. Therefore, the existence of EB would not completely prevent the oxidation of MEB completely, but suppress the diffusion of O₂ to a high extent in MEB composite, lowering the amount of O₂ around Ti₃C₂T_x.

As we mentioned in our manuscript, both oxygen and water molecules are responsible for the oxidation of the Ti₃C₂T_x (*Chem. Mater.* 2017 29, 4848-4856; *Inorg.*

Chem. 2019, 58, 1958-1966; *ACS Nano* 2021, 15, 5249-5262), in which O₂ is the key factor for the oxidation of Ti₃C₂T_x in the atmospheric environment. Due to the weaker oxidational capability of water molecules compared to oxygen molecules, TiO₂ occurs in Ti₃C₂T_x-RH 90%-400C-4 (Fig. R3-2) while it is quite obvious in Ti₃C₂T_x-air-400C-2 (Fig. R3-1). This could explain why there is no obvious TiO₂ peaks in MEB even after treated in RH 90% for 6h (Fig. R3-2). Based on the above analysis, we conclude that the oxidation of Ti₃C₂T_x could be suppressed to a very high extent in MEB composite, especially in humid environment.

Fig. R3-1 (a) Raman spectra of Ti₃C₂T_x films before and after treatment under synthetic air at 400 °C for 2, 4, 6 hours. (b) Raman spectra of MEB films before and after treatment under synthetic air at 400 °C for 2, 4, 6 hours. (**Fig. 2e, f** in revised manuscript)

Fig. R3-2 (a) Raman spectra of Ti₃C₂T_x films before and after treatment under 90% RH Ar at 400 °C for 2, 4, 6 hours. (b) Raman spectra of fresh MEB films before and after treatment under 90% RH Ar at 400 °C for 2, 4, 6 hours. (**Fig. S21** in revised manuscript)

Comment 2. Mechanical characterization of the membrane before and after the treatments is necessary for completeness. The provided optical images of folded specimens are not sufficient for assessing the mechanical integrity.

Response: Thanks for the reviewer’s comments. According to the reviewer’s suggestion, we have made some supplements.

As is shown in Fig. R3-3, pristine MEB and MEB-Air-400C-2 films perform good integrity under various deformations, such as bending and rolling, indicating the retention of flexibility even after high-temperature treatment. In addition, after ultrasonication for 1 min in DI water (Fig. R3-4), MEB-Air-400C-2 reserves the integrity. By contrast, $Ti_3C_2T_x$ tends to be fragile after annealing in air at 400 °C for 2h (Fig. R3-5) and turns to blue color because of the degradation of $Ti_3C_2T_x$. The above tests and analysis validate the mechanical integrity and flexibility of MEB even after high-temperature treatment in air.

Action: We have added the related figure in our revised Supplementary Information and added the analysis in the revised manuscript at page 4:

“As shown in Fig. 2b and Fig. S5, MEB-Air-400C-2 reserves the flexibility of pristine MEB and keeps its integrity after ultrasonication for 1 min in DI water (Fig. S6b).”

Fig. R3-3 Optical images showing the appearance and flexibility of pristine MEB and MEB-Air-400C-2 films (bending and rolling). (**Fig. S5** in revised manuscript)

Fig. R3-4 Photographs of the films annealed in synthetic air at 400 °C for 2 hours, and be treated by ultrasonication for 1min in water: (a) $\text{Ti}_3\text{C}_2\text{T}_x$ -Air-400C-2 (b) MEB-Air-400C-2. (**Fig. S6** in revised manuscript)

Fig. R3-5 Optical images of pristine $\text{Ti}_3\text{C}_2\text{T}_x$ film and MEB film before and after annealed at 400°C for 2 hours in air. (**Fig. 2b** in revised manuscript)

Comment 3. Information about the oxygen/ water barrier properties of EB should be given in order to support better the mechanism of antioxidation in MEB (e.g. experimental data or from literature).

Response: We appreciate the reviewer for pointing this out. XRD and SEM of EB are performed based on the same treatments with MEB (annealed at 400 °C for 6h and 600°C for 2h respectively in RH 90% synthetic air) in our manuscript. XRD patterns (Fig. R3-6) companied with SEM (Fig. R3-7) of EB-RH 90% air-400C-6 show that layered structure of EB retains without appearance of SiO_2 and Al_2O_3 after annealing. Furthermore, the lamellar structure retains with the reduction of the interlayer (d -) spacing, characterized by the peak of (002) shifting from $2\theta = 6.0^\circ$ to $\sim 8.0^\circ$ after

annealing at 600°C for 2 h. In summary, EB is thermally stable during the annealing with the presence of O₂/H₂O.

Furthermore, we displayed the thermal stability of EB by thermal gravimetric (TG) curves with mass spectrometry analysis (MS) in air and Ar, as shown in Fig. R3-6 and Fig. R3-7. As a result, only H₂O is detected from mass spectrum in the heating process (25 °C-1050 °C) under air and Ar atmospheres, suggesting the high thermal stability of EB. The degradation of EB starts at 700 °C because of the dissociation of -OH and =O terminations (*C. R. Chimie* 2018, 21, 391-398).

Action: We have added the XRD, SEM images of treated EB and relevant analysis in the revised Supplementary Information. Also, we made a modification in the revised manuscript at page 6:

“For EB, as shown in Fig. 3b and S10b, only H₂O is detected from mass spectrum in the heating process under air and Ar atmospheres, suggesting the high thermal stability of EB with further confirmed by XRD and SEM analyses in Fig. S11 and S12.”

Fig. R3-6 XRD patterns of oxidized EB and EB after annealing under different conditions. (**Fig. S11** in revised manuscript)

Fig. R3-7 SEM images of the cross-section of different samples: (a) EB-RH 90% air-400C-6; (b) EB-RH 90% air-600C-2. (**Fig. S12** in revised manuscript)

Fig. R3-8 Thermal gravimetric (TG) curves in air with mass spectrometry analysis (MS) for the atomic mass unit (amu) of 18/H₂O and 44/CO₂ for EB. (**Fig. 3b** in revised manuscript)

Fig. R3-9 Thermal gravimetric (TG) curves in Ar with mass spectrometry analysis (MS) for the atomic mass unit (amu) of 18/H₂O and 44/CO₂ for EB. (**Fig. S10b** in revised manuscript)

Comment 4. In the comparison of EMI SE of MEB with other materials in the literature (Table S1 and Figure 5f), the thickness of the shield cannot be neglected since the degree of shielding depends on it. Furthermore, the authors should provide further analysis of the shielding mechanism, based on theoretical predictions of different shielding contributions (i.e. reflection, adsorption, multiple reflection). The authors are therefore suggested to read the following references on these aspects: *Adv.Mater.*2013, 25, 1296–1300; *Nature Communications* 12, 4655 (2021); *Science* 353, 1137-1140 (2016).

Response: We thank the reviewer’s suggestions, and have made several amendments accordingly. The thickness and specific SE of the shielding materials are added in Table S1. Also, the specific SEs of different shielding materials are displayed in Fig. R3-10. Furthermore, we propose the shielding mechanism of MEB in our manuscript, based on theoretical predictions of different shielding contributions (i.e. reflection, adsorption, multiple reflection) and suggested literatures as shown in Fig R3-11. The EMI shielding originates from the high electrical conductivity of $Ti_3C_2T_x$ since EB is noneffective for THz shielding as shown in Fig. R3-12. As EM waves strike the surface of a MXene flake, some EM waves are immediately reflected (reflected waves) because of the highly conductive MXene (*Science* 2016, 353, 1137-1140). The remaining waves pass through the flack inducing a loss in energy by interacting with MXene, which is marked as absorbed waves. Because of the multiple reflections among MXene flakes, the intensity of EM waves decreased gradually. As a result, the total EMI shielding effectiveness (EMI SE) is the sum of the effectiveness of all attenuating mechanisms, including absorption, reflection, and the multiple reflections (*Adv. Mater.* 2013, 25, 1296-1300; *Nat. Commun.* 2021, 12, 4655).

Action: (a) The thickness and specific SEs of the shielding materials are added in Table S1. Also, the comparison of specific SE and working temperature for various THz shielding materials is displayed in Fig. S30 and be described at page 12:

“So far, most THz shielding materials are utilized at room temperature and few researchers focus on their applications at high temperature or hash environment, as shown in Fig. S30c. Therefore, MEB with high-temperature resistant property in air

and humid environment may meet the demands of working in harsh environment (further high-temperature THz test is necessary to finally evaluate the feasibility of MEB for practical scenario).”

(b) The proposed EMI shielding mechanism of MEB is added in the revised Supplementary Information (Fig. S29), then explained in the figure caption and manuscript at page 12:

“**Fig. S29** Proposed EMI shielding mechanism of MEB, based on theoretical predictions of different shielding contributions (i.e. reflection, adsorption, multiple reflection). The EMI shielding originates from the high electrical conductivity of $Ti_3C_2T_x$ since EB is noneffective for THz shielding. Each time the intensity of an EM wave is decreased when transmitted through a MXene flake. As a result, the total EMI shielding effectiveness (EMI SE) is the sum of the effectiveness of all attenuating mechanisms, including absorption, reflection, and the multiple reflections.”

“In order to truly reflect the performance comparison, MEB and $Ti_3C_2T_x$ are controlled to obtain the same amount of 11 mg $Ti_3C_2T_x$, as the EMI shielding originates from the high electrical conductivity of $Ti_3C_2T_x$ and EB is noneffective for THz shielding (Fig. S28), which is further explained by the proposed EMI shielding mechanism of MEB in Fig. S29.”

(c) Reference

4. Pavlou, C. *et al.* Effective EMI shielding behaviour of thin graphene/PMMA nanolaminates in the THz range. *Nat. Commun.* **12**, 4655 (2021).
5. Chen, Z., Xu, C., Ma, C., Ren, W. & Cheng, H. M. Lightweight and flexible graphene foam composites for high-performance electromagnetic interference shielding. *Adv. Mater.* **25**, 1296-1300 (2013).
7. Shahzad, F. *et al.* Electromagnetic interference shielding with 2D transition metal carbides (MXene). *Science* **353**, 1137-1140 (2016).

Fig. R3-10 Comparison of specific SE and treating temperature for various THz shielding materials. Each symbol indicates a set of material category as follows: CA-Graphene composites (deep grey open rhombus), CNW/PMC (red filled circle), SWNT (blue filled triangle), SWNT/PVA (green filled triangle), MWNT (purple filled square), Graphene/Acrylic (orange filled triangle), MXene/graphene/PDMS (purple circle), Cu/Graphene (green half-filled square), Gr/Cu/Gr (blue open star), Cu (blue filled circle), PUS-Ni/MXene (orange filled circle), MSF (deep grey open square), MXene waterborne painting (red circle), EMB in this work (pink star). A detailed description of each data point is presented in Table S1. (**Fig. 5f** in revised manuscript)

Table S1 Comparison of treating temperature for various THz shielding materials.

Materials	Measured Band (THz)	Average SE (dB)	Thickness (μm)	Specific SE (dB/ μm)	Treating Temperature ($^{\circ}\text{C}$)	Ref.
CA-Graphene composites	0.5-0.75	40	40	1.0	25	4
CNW/PMC	0.57-0.63	40	70	0.57	90	5
SWNT	0.2-2.5	38	0.95	40.0	25	6
SWNT/PVA	1.25-2.1	20	300	0.07	25	7
MWNT	0.4-2.2	25	25	1.0	25	8
Graphene/Acrylic	0.5-0.75	36	25	1.44	25	9
MXene/graphene/PDMS	0.2-2.0	45.3	400-500	0.11	25	10
Cu/Graphene	0.1-1.0	56.7	~ 30	1.89	120	11
Gr/Cu/Gr	0.1-1.0	41	~ 30	1.37	120	11
Cu	0.1-1.0	27.5	30	0.92	120	11
PUS-Ni/MXene	0.1-2.2	42.7	800	0.05	25	12
MSF	0.2-2.0	51	200	0.255	25	13
MXene waterborne painting	0.2-1.6	50.5	38.3	1.32	25	14
MEB	0.2-1.3	47	11	4.27	25	This work
MEB	0.2-1.3	50	9.5	5.26	400	This work
MEB	0.2-1.3	48	9.0	5.33	600	This work

References in Table S1 are listed in the Supplementary Information.

Fig. R3-11 Proposed EMI shielding mechanism of MEB. (Fig. S29 in revised manuscript)

Fig. R3-12 THz SE in 0.2-1.3 THz of EB film. (Fig. S28 in revised manuscript)

Reviewer #4:

Comment 1. *The authors have done an excellent job by using wide spectra of experimental and modelling evidences for validating their hypothesis on improving the thermal and ambient stability of $Ti_3C_2T_x$ MXene through the composition with bentonite (EB) nanosheets. The data were very convincing that oxygen molecules are preferentially adsorbed on EB by composing EB sheets with MXene/EB (MEB) composite sheets. This study could open a new window for wider environmental applications of MXene composites in aqueous and high temperature conditions. Although the 2D/2D Ti_3C_2 MXene/bentonite (Bt) heterojunction was fabricate and investigated earlier for photocatalytic CO_2 reduction. <https://doi.org/10.1016/j.cej.2020.125868>, this study is still important to provide a guide to tailor the stability of MXenes to the certain environmental application. Specifically in this study the thermally treated MEB at temperatures up to 800 °C and different RH conditions was able to retain a very good THz shielding about 50 dB at 500 °C and 48 dB at 600 °C, as compared to pristine MXene which is know to be prone to severe degradation at this conditions.*

Response: We sincerely appreciate the reviewer's positive comments on our work. As the reviewer said, 2D/2D Ti_3C_2 MXene/bentonite (Bt) heterojunction was fabricated and investigated in precious study (<https://doi.org/10.1016/j.cej.2020.125868>), which provides a guide for our work. Therefore, we have studied this article carefully and cited in our manuscript.

Action: Reference

27. Tahir, M. & Tahir, B. 2D/2D/2D O- C_3N_4 /Bt/ $Ti_3C_2T_x$ heterojunction with novel MXene/clay multi-electron mediator for stimulating photo-induced CO_2 reforming to CO and CH_4 . *Chem. Eng. J.* **400**, 125868 (2020).

Comment 2. I have only one “Major” concern that the authors have done the shielding measurements at room temperature after exposing the MEB to different annealing conditions. I am wondering if this is enough to make the claim on the shielding performance at high temperatures.

in order to validate that MEB can be “promising for THz shielding at high temperature in oxidizing environment for a long operation time” I see it is crucial to conduct the Terahertz shielding measurements at elevated temperatures. It is well accepted that MXenes lamellar films show different swelling/contraction profile at different temperatures. And I assume this will have a strong impact on the shielding performance at these conditions, and not necessarily comparable with the performance at room temperature.

Response: We appreciate the reviewer for the constructive suggestion. High-temperature resistant electromagnetic interference (EMI) shielding is of importance for aerospace industry, including but not limited to the EMI shielding of engine casing of aircraft (higher than 380 °C). It is indeed important to conduct the terahertz shielding measurements at elevated temperatures. Nevertheless, suffering from the constraints of current terahertz testing setup, terahertz shielding test of the samples cannot be conducted at high temperature in real time. Thus, the project of measuring the terahertz shielding efficiency of our samples at different elevated temperatures could not be implemented so far. To eliminate the misunderstanding, we have revised the sentence “promising for THz shielding at high temperature in oxidizing environment for a long operation time” to be “therefore making MEB being promising high-temperature resistant THz shielding materials.” In fact, more efforts would be put into exploring the hash-environment adaptive testing setup of terahertz electromagnetic waves in our future work.

Though we are unable to conduct real-time high temperature shielding test, we have tested the EMI shielding performance changes of the samples before and after annealing treatment. The annealing process of MEB was conducted in atmosphere with the existence of O₂ or/ and H₂O at 400°C ~600°C (Table R4-1). The nearly unchanged THz shielding performance of MEB in 0.2-1.3 THz, as shown in Fig. R4-1, proves its good

stability at high temperature. Therefore, it is reasonable to believe that MEB with high-temperature resistant property in air and humid environment may meet the demands of working in high-temperature scenario.

Action: we has revised the sentence “promising for THz shielding at high temperature in oxidizing environment for a long operation time” as followed, at page 12:

“All the results suggest that the introducing of EB can suppress the oxidation-induced deterioration of $Ti_3C_2T_x$, therefore making MEB being promising high-temperature resistant THz shielding materials.”

Table R4-1 The list of annealing samples in our work

Samples	Atmosphere	Temperature (°C)	Annealing time(h)	
$Ti_3C_2T_x$	1	Synthetic air	400	2
	2	Synthetic air	400	4
	3	Synthetic air	400	6
	4	RH 90% Ar	400	2
	5	RH 90% Ar	400	4
	6	RH 90% Ar	400	6
	7	Atoms	400	6
	8	Atoms	500	2
	9	Atoms	600	2
MEB	1	Synthetic air	400	2
	2	Synthetic air	400	4
	3	Synthetic air	400	6
	4	RH 90% Ar	400	2
	5	RH 90% Ar	400	4
	6	RH 90% Ar	400	6
	7	Atoms	400	6
	8	Atoms	500	2
	9	Atoms	600	2

Synthetic air with volume fractions of 21% O_2 and 79% N_2 . Atoms: RH 60% synthetic air (simulated atmospheric environment in practical scenario).

Fig. R4-1. (a) EMI SE in 0.2-1.3 THz of $\text{Ti}_3\text{C}_2\text{T}_x$ -RT, MEB-RT, $\text{Ti}_3\text{C}_2\text{T}_x$ -Atmos-400C-6, MEB-Atmos-400C-6. (b) EMI SE in 0.2-1.3 THz of $\text{Ti}_3\text{C}_2\text{T}_x$ -Atmos-500C-2, 600C-2, and MEB-Atmos-500C-2, 600C-2. (c) The average THz SE of the samples showed in Fig. R4-1a (left) and Fig. R4-1b (right). (**Fig. 5** in revised manuscript)

Reviewer #5

Comment 1. *In this manuscript, Air-stable $Ti_3C_2T_x$ based composite for high-temperature resistant terahertz shielding, has been investigated. The effect of air on the heating of Ti_3C_2 MXenes, and their further testing are investigated in detail. However, this manuscript missing important investigation related to MXene and should be further tested. Some comments are:*

It is well known that MXenes has functional group over its surface, is there any role of MXene with bentonite to improve high-temperature resistant terahertz.

Response: We thank the reviewer for this constructive suggestion. In general, the surface termination affects the chemical and physical properties of MXene. Therefore, to validate the influence of termination on the final high-temperature resistant property, we prepared $Ti_3C_2T_x$ through two approaches which shows different termination ratios: **HF- $Ti_3C_2T_x$:** $Ti_3C_2T_x$ was synthesized by selectively etching Al of Ti_3AlC_2 MAX using HF and HCl (*Adv. Energy Mater.* 2020, 11; *ACS Nano* 2021,15, 13646–13652). 1 g of Ti_3AlC_2 powder (11 Technology Co.) was added to the etchant solution, which includes 12 mL 38 wt.% hydrochloric acid (HCl, Sigma-Aldrich), 2 mL 49 wt.% hydrofluoric acid (HF, Sigma-Aldrich) and 6 mL deionized (DI) water. Next, the Ti_3AlC_2 powder was etched for 24 h by string at 400 rpm at 35 °C. Then, the multilayered $Ti_3C_2T_x$ was obtained by washing the reaction mixture with DI water via centrifugation at 3500 rpm (5 min per cycle). After washing for 5 ~ 6 times (until the pH \geq 6), the sediment was dispersed into the delamination solution, which was prepared by adding 1 g of LiCl (Sigma-Aldrich) into 40 mL DI water. Then, the mixture was stirred for 4 h at room temperature. The dispersion was washed using DI water and the single layer $Ti_3C_2T_x$ aqueous solution was obtained by centrifugation at 3500 rpm for 5 min.

MILD- $Ti_3C_2T_x$: LiF (1.6 g; Sigma-Aldrich) was added to 9 M HCl (20 ml; HCl, 38wt.%, Sigma-Aldrich) and continuously stirred at 400 rpm for 30 minutes at 40 °C. 1g Ti_3AlC_2 (11 Technology Co.) was added into the mixture solution and stirred at 400 rpm for 24 h at 40 °C. Then the dispersion was washed by DI water for 4 ~ 6 times until the pH \geq 6. After washing, the sediment was dispersed in 50 mL fresh DI water and the

single layer $\text{Ti}_3\text{C}_2\text{T}_x$ aqueous solution was obtained by centrifugation at 3500 rpm for 5 min (*Adv. Mater.* 2020, 32, e2001093).

The termination analysis of HF- and MILD- $\text{Ti}_3\text{C}_2\text{T}_x$: The termination analysis of HF- $\text{Ti}_3\text{C}_2\text{T}_x$ and MILD- $\text{Ti}_3\text{C}_2\text{T}_x$ (characterized by SEM and XRD, Fig. R2-4) was conducted through XPS as shown in Fig. R5-2. By fitting the relative intensities of the O 1s, F 1s, Cl 2p peak (*ACS Nano* 2019, 13, 9449-9456; *Chem. Mater.* 2019, 31, 6590-6597), we determine the termination of HF- $\text{Ti}_3\text{C}_2\text{T}_x$ MXene (T_x) to be 38.7% =O, 31.6% -F, 7.7% -Cl, 22.0% -OH and MILD- $\text{Ti}_3\text{C}_2\text{T}_x$ with 43.8% =O, 24.4% -F, 13.1% -Cl, 18.7% -OH, which reveals the different termination ratios for two samples.

Characterization of pristine MILD $\text{Ti}_3\text{C}_2\text{T}_x$ and MILD MEB (MILD $\text{Ti}_3\text{C}_2\text{T}_x$ /EB composite) after high-temperature annealing in oxygen atmosphere: We investigated the antioxidation of MILD $\text{Ti}_3\text{C}_2\text{T}_x$ and MILD MEB by the same treatment with $\text{Ti}_3\text{C}_2\text{T}_x$ and MEB (displayed in our original manuscript). Similarly, the samples were annealed at 400°C under synthetic air (with volume fractions of 21% O_2 and 79% N_2) for different time, which are denoted as MILD $\text{Ti}_3\text{C}_2\text{T}_x$ /MEB-Air-400C-t (t is annealing time, hours). The cross-sectional SEM images (Fig. R5-3) show a good retention of layered structure for MILD MEB-Air-400C-2, 4, 6. At the same time, TiO_2 particle is obvious in MILD $\text{Ti}_3\text{C}_2\text{T}_x$ -Air-400C-2, 4, 6, and the degrading process is accelerated by elevated temperature. The oxidation of different samples is further confirmed by Raman spectra displayed in Fig. R5-4. Typical E_g (Ti, C) mode at 289, 369 cm^{-1} and A_{1g} (Ti, C) mode at 201, 599, 722 cm^{-1} of $\text{Ti}_3\text{C}_2\text{T}_x$ remain high strength in MILD MEB-Air-400C-2 and A_{1g} peak (201 cm^{-1}) is still obvious in MILD MEB-Air-400C-6. By contrast, MILD $\text{Ti}_3\text{C}_2\text{T}_x$ -Air-400C-2 shows the transformation of Ti-C bonding to Ti-O bonding, which is indicated by the appearance of TiO_2 peaks at 145, 400, 519, 637 cm^{-1} . It confirms the suppression of oxidation for MILD MEB at high temperature in air. In summary, MEB and MILD MEB with different termination ratios on $\text{Ti}_3\text{C}_2\text{T}_x$ display similar high-temperature resistant property in oxygen atmosphere. It reveals that the terminations on $\text{Ti}_3\text{C}_2\text{T}_x$ may have little influence on the high-temperature resistant property of the MEB composite.

Action: We have added the synthesis method of MILD MEB and relevant analysis in the revised manuscript, page 10-11. In addition, we added the related figures and the analysis to the revised Supplementary Information.

“The effect of termination ratio on high-temperature resistant property of MEB.

To detect the influence of terminations on the high-temperature resistant property of $Ti_3C_2T_x$, two approaches are adopted to synthesize $Ti_3C_2T_x$, which are minimally intensive layer delamination method (denoted to MILD- $Ti_3C_2T_x$, Fig. S25) and HF etching method (HF- $Ti_3C_2T_x$), respectively. By fitting the relative intensities of the O 1s, F 1s, Cl 2p peaks^{36,37} as shown in Fig. 4a-d and S26, we determine the termination of HF- $Ti_3C_2T_x$ to be 38.7% =O, 31.6% -F, 7.7% -Cl, 22.0% -OH and MILD- $Ti_3C_2T_x$ with 43.8% =O, 24.4% -F, 13.1% -Cl, 18.7% -OH, which reveals the different termination ratios for two samples. For clear description, composite consisting of MILD- $Ti_3C_2T_x$ and EB is denoted as MILD MEB in order to differentiate with MEB (HF- $Ti_3C_2T_x$ /EB). Following the same thermal treatment of MEB, MILD MEB-Air-400C-2 shows high-intensive E_g (Ti, C) peaks at 289, 369 cm^{-1} and A_{1g} (Ti, C) peaks at 201, 599, 722 cm^{-1} of $Ti_3C_2T_x$, and A_{1g} peak (201 cm^{-1}) is still obvious in MILD MEB-Air-400C-6 as shown in Fig. 4f. By contrast, MILD $Ti_3C_2T_x$ -Air-400C-2 (Fig. 4e) shows the transformation of Ti-C bonding to Ti-O bonding, which is indicated by the appearance of TiO_2 peaks at 145, 400, 519, 637 cm^{-1} . Along with the SEM analysis (Fig. S27), the results confirm the high-temperature resistant property for MILD MEB in air. The similar behavior in MEB and MILD MEB reveals our strategy on suppressing the oxidation of $Ti_3C_2T_x$ is independent on the termination ratio of $Ti_3C_2T_x$.”

“MILD- $Ti_3C_2T_x$ (prepared by minimally intensive layer delamination method): LiF (1.6 g; Sigma-Aldrich) was added to 9 M HCl (20 ml; HCl, 38wt.%, Sigma-Aldrich) and continuously stirred at 400 rpm for 30 minutes at 40 °C. 1g Ti_3AlC_2 (11 Technology Co.) was added into the mixture solution and stirred at 400 rpm for 24 h at 40 °C. Then the dispersion was washed by DI water for 4 ~ 6 times until the pH \geq 6. After washing, the sediment was dispersed in 50 mL fresh DI water and the single layer $Ti_3C_2T_x$ aqueous solution was obtained by centrifugation at 3500 rpm for 5 min⁴⁰.”

“Meanwhile, MILD $\text{Ti}_3\text{C}_2\text{T}_x$ film and MILD MEB are fabricated by MILD $\text{Ti}_3\text{C}_2\text{T}_x$ solution in the similar process.”

Reference:

36. Chen, X. *et al.* N-Butyllithium-treated $\text{Ti}_3\text{C}_2\text{T}_x$ MXene with excellent pseudocapacitor performance. *ACS Nano* **13**, 9449-9456 (2019).
37. Schultz, T. *et al.* Surface termination dependent work function and electronic properties of $\text{Ti}_3\text{C}_2\text{T}_x$ MXene. *Chem. Mater.* **31**, 6590-6597 (2019).
40. Zhang, J. *et al.* Scalable manufacturing of free-standing, strong $\text{Ti}_3\text{C}_2\text{T}_x$ MXene films with outstanding conductivity. *Adv. Mater.* **32**, e2001093 (2020).

Fig. R5-1. (a) SEM image of MILD- $\text{Ti}_3\text{C}_2\text{T}_x$ flake; (b) XRD pattern of MILD- $\text{Ti}_3\text{C}_2\text{T}_x$. The results show typical nanosheet morphology with lateral size of $\sim 12 \mu\text{m}$ for $\text{Ti}_3\text{C}_2\text{T}_x$. (**Fig. S25** in revised manuscript)

Fig. R5-2. (a-c) XPS of O 1s, F 1s and Cl 2p signals, respectively, for the pristine HF-Ti₃C₂T_x and MILD-Ti₃C₂T_x films. (d) XPS survey spectra of HF-Ti₃C₂T_x and MILD-Ti₃C₂T_x films with the ratio of O, F, Cl as a elements list (inserted). (e) The termination ratio of HF-Ti₃C₂T_x and MILD-Ti₃C₂T_x measured by XPS with a table list (inserted). (**Fig. 4a-d and Fig. S26** in revised manuscript)

Fig. R5-3. SEM images of the cross-section of different samples: (a1) MILD $\text{Ti}_3\text{C}_2\text{T}_x$ -Air-400C-2, (b1) MILD $\text{Ti}_3\text{C}_2\text{T}_x$ -Air-400C-4, (c1) MILD $\text{Ti}_3\text{C}_2\text{T}_x$ -Air-400C-6; (a2) MILD MEB-Air-400C-2, (b2) MILD MEB-Air-400C-4, (c2) MILD MEB-Air-400C-6. (Fig. S27 in revised manuscript)

Fig. R5-4. (a) Raman spectra of MILD $\text{Ti}_3\text{C}_2\text{T}_x$ before and after treatment under synthetic air at 400 °C for 2, 4, 6 hours. (b) Raman spectra of MILD MEB before and after treatment under synthetic air at 400 °C for 2, 4, 6 hours. (Fig. 4e-f in revised manuscript)

Comment 2. It has been reported that MXene was converted to TiO_2 by heating. Is there any critical role of TiO_2 to control high temperature? If it is true, then more results and such testing should be conducted. For example, presently, role of TiO_2 is critically discussed and suggested articles are *Journal of Colloid and Interface Science*, 2021, 591, pp. 20–37; and *Applied Catalysis B: Environmental*, 2021, 285, 119777

Response: In previous research, thermal annealing of $\text{Ti}_3\text{C}_2\text{T}_x$ films at elevated temperatures ($\sim 600^\circ\text{C}$) for 30 min causes the formation of a protective layer of TiO_2 on the outermost layer of films, which suppresses the oxidizing process in aqueous solutions at room temperature to some extent (*ACS Appl. Nano Mater.* 2020, 3, 10578-10585). Nevertheless, the thickness of TiO_2 particles membrane increased with the prolonging of annealing time in air as shown in Fig. R5-6. According to our experiment that treating the samples at high temperature in air for more than 2 hours, the initially generated TiO_2 particles is less effective for suppressing O_2 diffusion and hence cannot decelerate the oxidation of $\text{Ti}_3\text{C}_2\text{T}_x$ to TiO_2 . In our work, the appearance and the amount of TiO_2 in the annealing samples reflect the degree of the oxidation of $\text{Ti}_3\text{C}_2\text{T}_x$. The large amount of TiO_2 particles observed in SEM images and the high intensities of TiO_2 peaks in Raman spectra reveal severe oxidation of $\text{Ti}_3\text{C}_2\text{T}_x$. The suggested articles have discussed the formation process and the role of TiO_2 which provide a guide for us to better understand the stability of MXene. Thus, we add the suggested articles in our references.

Action: Reference

30. Khan, A. A. & Tahir, M. Well-designed 2D/2D $\text{Ti}_3\text{C}_2\text{T}_{A/R}$ MXene coupled g- C_3N_4 heterojunction with in-situ growth of anatase/rutile TiO_2 nucleates to boost photocatalytic dry-reforming of methane (DRM) for syngas production under visible light. *Applied Catalysis B: Environmental* **285**, (2021).
31. Tahir, M. & Tahir, B. In-situ growth of TiO_2 imbedded $\text{Ti}_3\text{C}_2\text{T}_A$ nanosheets to construct PCN/ $\text{Ti}_3\text{C}_2\text{T}_A$ MXenes 2D/3D heterojunction for efficient solar driven photocatalytic CO_2 reduction towards CO and CH_4 production. *J. Colloid Interface Sci.* **591**, 20-37 (2021).

Fig. R5-6 SEM images of the cross-section of different samples: (a) $\text{Ti}_3\text{C}_2\text{T}_x\text{-Air-400C-2}$; (b) $\text{Ti}_3\text{C}_2\text{T}_x\text{-Air-400C-4}$; (c) $\text{Ti}_3\text{C}_2\text{T}_x\text{-Air-400C-6}$. (**Fig. S8** in revised manuscript)

Comment 3. Author should add comparative analysis of MAX and MXene at different temperatures to see the critical role of either morphology, structure, functional group or any other factors and some suggested article is *Journal of Alloys and Compounds*, 2020, 842, 155752

Response: According to the reviewer's suggestions, the analysis of MAX has been conducted. As is shown in Fig. R5-7, the Ti_3AlC_2 particle ($\approx 500 \mu\text{m}^3$) is predominately made up of square, platelet-like particles, which is consistent with the literatures report (*Adv. Mater.* 2011, 23, 4248-4253; *ACS Nano* 2021, 15, 6420-6429; *J. Alloys Compd.* 2020, 842). As reported previously, Ti_3AlC_2 is resistant to oxidation in air under 900-1400 °C (*Materials Research Letters* 2013, 1, 115-125; *Corros. Sci.* 2003, 45, 891-907). Beyond 1400 °C, TiO_2 and Al_2O_3 layers will be formed in the oxidizing progress of Ti_3AlC_2 , which can be manipulated into the following equation:

As is shown Fig. R5-8, the XRD patterns of $\text{Ti}_3\text{AlC}_2\text{-RT}$ and $\text{Ti}_3\text{AlC}_2\text{-Atmos-600C-2}$ (annealing under 60% RH air at 600°C for 2h) show the similar peaks with high intensity of (002) (011) (012) (014) (015) and (110), which confirms the stability of Ti_3AlC_2 at 600°C. By contrast, $\text{Ti}_3\text{C}_2\text{T}_x$ is degraded to TiO_2 completely after annealing under 60% RH air at 600°C for 2h. It reveals that Ti_3AlC_2 is more capable in resistance of oxidation than pure $\text{Ti}_3\text{C}_2\text{T}_x$.

Action: The SEM and XRD analyses of Ti_3AlC_2 are added in the revised supplementary information (Fig. S31-S32).

Reference

38. Tasleem, S., Tahir, M. & Zakaria, Z. Y. Fabricating structured 2D Ti_3AlC_2 MAX dispersed TiO_2 heterostructure with Ni_2P as a cocatalyst for efficient photocatalytic H_2 production. *J. Alloys Compd.* **842**, 155752 (2020).

Fig. R5-7 SEM image of a square Ti_3AlC_2 particle. (**Fig. S31** in revised manuscript)

Fig. R5-8 (a) XRD patterns of $\text{Ti}_3\text{AlC}_2\text{-RT}$ and $\text{Ti}_3\text{AlC}_2\text{-Atmos-600C-2}$; (b) XRD patterns of $\text{Ti}_3\text{C}_2\text{T}_x\text{-RT}$ and $\text{Ti}_3\text{C}_2\text{T}_x\text{-Atmos-600C-2}$. (**Fig. S32 and S24** in revised manuscript)

Comment 4. *When MXene was mixed with Bentonite, what was the role of TiO₂, should be discussed (Chemical Engineering Journal, 2020, 400, 125868)*

Response: In our work, Ti₃C₂T_x MXene/EB was prepared and went through the treatment of annealing at 400 °C, 600°C in air for more than 2 hours, through which, the Ti₃C₂T_x nanosheets degraded to TiO₂ particles inducing defects for compact MEB lamellar structure. Therefore, the initially generated TiO₂ particles is less effective for suppressing O₂ diffusion and hence cannot decelerate the oxidation of Ti₃C₂T_x to TiO₂. The appearance and the amount of TiO₂ in the annealing samples reflect the degree of the oxidation of Ti₃C₂T_x. The large amount of TiO₂ particles observed in SEM images and the high intensities of TiO₂ peaks in Raman spectra reveal sever degradation of Ti₃C₂T_x. The suggested articles have added in the references of our revised manuscript.

Action: Reference

27. Tahir, M. & Tahir, B. 2D/2D/2D O-C₃N₄/Bt/Ti₃C₂T_x heterojunction with novel MXene/clay multi-electron mediator for stimulating photo-induced CO₂ reforming to CO and CH₄. *Chem. Eng. J.* **400**, 125868 (2020).

Comment 5. More critical analysis about the materials and their role should be discussed throughout the manuscript.

Response: We appreciate the efforts the reviewer spent on our work. The critical analyses about the materials in our work have been made and listed in Table R5-1:

Table R5-1 The critical analysis about the materials in our work

Material	Analysis (Fig.)								
	SEM	TEM/STEM-EDX	XRD	XPS	Raman	TG	FT-IR	Mechanical Integrity	Zeta potential
Ti₃AlC₂	Fig.S31		Fig.S32						
	Fig.2								
	Fig.S2								
	Fig.S4		Fig.S1	Fig.3					
Ti₃C₂T_x	Fig.S8	Fig.2	Fig.S9	Fig.4	Fig.2	Fig.3		Fig.2	
	Fig.S23	Fig.S2	Fig.S24	Fig.S19	Fig.4	Fig.S10	Fig.S20	Fig.S6	Fig.S3
	Fig.S24		Fig.S25	Fig.S26	Fig.S21				
	Fig.S25								
	Fig.S27								
EB	Fig.S2		Fig.S1			Fig.3			
	Fig.S4	Fig.S2	Fig.S11			Fig.S10	Fig.S20		Fig.S3
	Fig.S12								
MEB	Fig.2							Fig.2	
	Fig.S4								
	Fig.S7	Fig.2	Fig.S9	Fig.3	Fig.2	Fig.3		Fig.2	
	Fig.S22		Fig.S24	Fig.S19	Fig.4		Fig.S20	Fig.S5	Fig.S3
	Fig.S24				Fig.S21			Fig.S6	
	Fig.S27								
h-BN	Fig.S15		Fig.S15		Fig.S18				
MBN	Fig.S17				Fig.S18				

The characterizations of materials we added in our revised manuscript are marked in red in Table R5-1

Discussions on the materials according to the above characterizations

(1) Ti₃AlC₂:

The Ti₃AlC₂ particle ($\approx 500 \mu\text{m}^3$) is predominately made up of square, platelet-like particles, which is consistent with previous literatures (*Adv. Mater.* 2011, 23, 4248-4253).

(2) Ti₃C₂T_x:

In the previous manuscript, we only used HF method to synthesize Ti₃C₂T_x (HF-Ti₃C₂T_x). Since different synthesis process may bring different ratio of terminations on Ti₃C₂T_x, we would like to validate if our strategy of fabricating high-temperature resistant composite is applicable on Ti₃C₂T_x synthesized through other methods. To do it, we used MILD (minimally intensive layer delamination) method to synthesize Ti₃C₂T_x with different termination ratios compared to Ti₃C₂T_x obtained from HF method (HF-Ti₃C₂T_x MXene with 38.7% =O, 31.6% -F, 7.7% -Cl, 22.0% -OH and MILD-Ti₃C₂T_x with 43.8% =O, 24.4% -F, 13.1% -Cl, 18.7% -OH). As a result, MILD-Ti₃C₂T_x exhibits the similar oxidation process compared with HF-Ti₃C₂T_x, indicating that the different termination ratios have little influence on the oxidation of Ti₃C₂T_x under O₂ atmosphere at high temperature.

(3) Extracted bentonite (EB):

The synthesized EB nanosheet has abundant hydroxy termination. EB displays good heat resistant property, and is stable below 700 °C. The results further confirm EB with thermal stability plays an important role in the composite (Ti₃C₂T_x MXene/EB).

(4) Ti₃C₂T_x MXene/EB (MEB)

According to STEM-EDX, the orientation and restacking nature of Ti₃C₂T_x and EB were shown clearly. This could further support our explanation on the mechanism of high-temperature resistant property in MEB which is speculated as: i) the more compact layered structure of MEB; ii) the effect of EB on the interaction between adsorbed O₂ and Ti₃C₂T_x. Moreover, although the termination ratios on Ti₃C₂T_x with different synthesis methods were not identical, their composites with EB showed similar high-temperature resistant property, demonstrating the universality of our strategy on suppressing the oxidation of Ti₃C₂T_x at high temperature under O₂ atmosphere.

(5) *h*-BN

h-BN is 2D nanosheet with good heat retardant property, which can be a contrast sample to detect the mechanism of high-temperature antioxidation in MEB.

(6) MXene/*h*-BN (MBN)

Compared with MEB, MBN exhibits poor antioxidation at high-temperature under O₂

atmosphere. Since both of *h*-BN and EB exhibit good heat retardant property, the different thermal stability performance of their composites with $\text{Ti}_3\text{C}_2\text{T}_x$ excludes the possibility that the high-temperature resistance of MEB comes from the heat retardant property of EB.

In summary, we design the experiment properly in our revised manuscript. Supplementary characterizations of samples (listed in Table R5-1) were conducted carefully and related analysis was added in the revised manuscript. The new results further confirm the role of EB on suppressing the oxidation process of $\text{Ti}_3\text{C}_2\text{T}_x$ at high temperature with the presence of O_2 . Furthermore, utilized as THz shielding materials, the antioxidized MEB performs high THz EMI SE of 48 dB after annealing under 600 °C in atmospheric environment, which validates MEB a promising candidate for THz shielding material that requires high-temperature working scenario. Meanwhile, we demonstrate the potential application of MEB in Joule heating and thermal-cycling, which is potentially suitable for thermal/photothermal catalysis and electric/heat responsive actuator.

REVIEWER COMMENTS

Reviewer #1 (Remarks to the Author):

Authors have incorporated all raised issues carefully which has caused to improve the quality of manuscript significantly. Therefore, the revised manuscript can be accepted in the current form.

Reviewer #2 (Remarks to the Author):

Thank you for coming back on my original comments. In general the actions are well received and they definitely improve the quality of the paper. I have listed below additional comments on your revision. However, I feel that the work still lacks novelty in the field despite it demonstrating the applicability of MXene and therefore would not be suitable for nature communications. In summary, the quality and quantity of the work is not in question.

1. In regards to the joule heating trials, it would be useful to understand more about the thermal properties of the material. In addition, the author's state a very high electrical conductivity for their pristine mxene film. Could you comment in more detail on how the joule heating effect was achieved? It would be ideal to know the power supplied in relation to temperature recorded. Finally, efficiency of the MEB composite as a joule heating material must be compared with others.
2. Is the extracted bentonite the only option? Are there any other group of materials that can offer these properties?
3. Typically the oxidation of mxene flakes starting from the edge, but figure 1 the schematic figure seems to show oxidation occurring in between layers. Could the authors correct this image.
4. In figure S3, its mentioned "Fig. S3 (a) Photograph of pristine Ti₃C₂T_x and the MEB aqueous dispersions (with 10, 30, 50, 70, 90 wt.% EB which are the dispersions to prepare MEB-0.1, MEB-0.3, MEB-0.5, MEB-0.7, MEB-0.9, respectively). " there are five kinds of composite. but in the main manuscript, can you clarify which sample is being tested.
5. How did u prepare the film in figure S4, in figure S4b, the SEM of MEB, which part is Mxene and which part is EB
6. Which EB composition are you using in S2
7. For figure 2c, What do u mean by thick chunk, which is it made of? And in the sem image, the top layers are showing very different deformation while the bottom part showing nothing, can u explain why?
8. In page 9, humidity was also considered as a factor, but the test environment is 400 degree in argon with humidity 90%. But this is not a normal environment. The testing environment should be typical oxygen/nitrogen environment (air) with humidity within 100 degree.
9. In figure S11, the xrd peaks seem to shift and broaden. It seems improper to make the deduction that the compositions disappear. The authors need to explain this phenomenon rather than stating that the the peaks are absent.
10. In figure S12, EB 400C-6 h and 600C-2h were compared. Temperature and duration are both changing, so it is hard to compare

Reviewer #3 (Remarks to the Author):

The authors have revised and restructured the manuscript by addressing some of reviewers' comments. Compared to the previous version, the authors, in fact, have not provided new information on the materials and their behaviour as was required by the reviewers, but only further explanation of existing results.

Specifically,

- Oxidation issue: I am satisfied with the response provided by the authors on the spectroscopic findings suggesting that oxidation is not completely prevented in MEB. EB is supposed to suppress (I would say "delay") the diffusion of O₂. However, the "further comment" I was expecting in the main text has not been added (even in the SI!). In this regard, clear/ direct information on the water/oxygen barrier properties (i.e. permeability!) of the system was not provided; therefore I am not fully satisfied with the response of the authors to comment 3.
- Mechanical integrity of the membranes before and after the treatment: as mentioned in the

previous round of revision, optical images of bent or deformed membranes are not sufficient in order to assess the mechanical integrity! Proper mechanical characterization results (e.g. stress/strain curves) are required for completeness and they are not provided in the revised version of the manuscript.

-EMI shielding : I am now satisfied with the analysis of the data based on the specific SE to compare with SOTA materials; however, theoretical predictions of the different shielding contributions (that were clearly required in my comment) was not provided in the revised manuscript. Without some theoretical predictions, the qualitative explanation of the shielding mechanism that has been added is not satisfactory.

Response Letter

(High-temperature resistant $Ti_3C_2T_x$ based composite in air)

We would like to express our gratitude to the editor and the reviewers for their valuable comments on our original and 1st revised manuscript. We have carefully revised our manuscript based on the reviewers' suggestions and the main changes are highlighted in yellow in the revised manuscript. The point-by-point response is attached as follows:

Reviewer #1

Authors have incorporated all raised issues carefully which has caused to improve the quality of manuscript significantly. Therefore, the revised manuscript can be accepted in the current form.

Response: We appreciate the efforts the reviewer has spent on our manuscript. Thanks a lot for all the positive comments on our work.

Reviewer #2

Thank you for coming back on my original comments. In general the actions are well received and they definitely improve the quality of the paper. I have listed below additional comments on your revision. However, I feel that the work still lacks novelty in the field despite it demonstrating the applicability of MXene and therefore would not be suitable for nature communications. In summary, the quality and quantity of the work is not in question.

Response: We appreciate the efforts the reviewer has spent on our manuscript. The point-by-point response has been done as follows:

Comment 1. *In regards to the joule heating trials, it would be useful to understand more about the thermal properties of the material. In addition, the author's state a very high electrical conductivity for their pristine mxene film. Could you comment in more detail on how the joule heating effect was achieved? It would be ideal to know the power supplied in relation to temperature recorded. Finally, efficiency of the MEB composite as a joule heating material must be compared with others.*

Response: Thanks for the reviewer's constructive comments. According to previous researches, $Ti_3C_2T_x$ MXene exhibits metallic conductivity (up to 2×10^4 S/cm) (ACS

Nano 2021, 15, 6420-6429) and high thermal conductivity [55.8 W/(m·K)] (*Sci. Adv.* 2019, 5, eaaw7956; *ACS Omega*, 2018, 3, 2609-2617), which meets the demand of Joule heating device on basis of Joule's law, $Q = I^2Rt = U^2t/R$ and $P = U^2/R$ where Q is the generated Joule heat, I is the DC current, U is the DC voltage, P is the output power of Joule heater, R is the resistance of film, t is the working time (*ACS Nano*, 2020, 14, 8793-8805).

In our work, the $Ti_3C_2T_x$ and MEB freestanding films were used to make the Joule heater device, in which the resistances of $Ti_3C_2T_x$ and MEB are 2 Ω and 6 Ω , respectively (Fig. R2-1). The heating performance (including steady temperature and heating rate) of the films was conducted under a driving voltage of 3.0 V as shown in Fig. R2-2. According to Joule's law, the heat is generated by the current flow through the films, which induced the temperature increase of the film. In our Joule heating devices, the output power of $Ti_3C_2T_x$ is 4.5 W and the output power of MEB is 1.5 W.

Furthermore, the comparison of the steady temperature and heating rate for various Joule heating materials was given in Table R2-1. The steady temperature of MEB is higher than MXene/aramid nonwoven fabric and MXene/cellulose fabric under the driving voltage of 3 V. Besides, the heating rate of MEB in our work is 20 $^{\circ}C s^{-1}$, much higher than graphene/MXene fabrics and PEDOT/PSS film shown in Table R2-1.

Action: We have added the Fig. S33 and Table S3 accompanied with the analysis in Supplementary Information:

“According to previous researches, $Ti_3C_2T_x$ MXene exhibits metallic conductivity (up to 2×10^4 S/cm) and high thermal conductivity [55.8 W/(m·K)], which meets the demand of Joule heating device on basis of Joule's law, $Q = I^2Rt = U^2t/R$ and $P = U^2/R$ where Q is the generated Joule heat, I is the DC current, U is the DC voltage, P is the output power of Joule heater, R is the resistance of film, t is the working time.

In our work, the $Ti_3C_2T_x$ and MEB freestanding films were used to make the Joule heater device, in which the resistances of $Ti_3C_2T_x$ and MEB are 2 Ω and 6 Ω , respectively (Fig. S33). The heating performance (including steady temperature and heating rate) of the films was conducted under a driving voltage of 3.0 V as shown in

Fig. 5d. According to Joule's law, the heat is generated by the current flow through the films, which induced the temperature increase of the film. In our Joule heating devices, the output power of $\text{Ti}_3\text{C}_2\text{T}_x$ is 4.5 W and the output power of MEB is 1.5 W.

Furthermore, the comparison of the steady temperature and heating rate for various Joule heating materials was given in Table S3. The steady temperature of MEB is higher than MXene/aramid nonwoven fabric and MXene/cellulose fabric under the driving voltage of 3 V. Besides, the heating rate of MEB in our work is $20\text{ }^\circ\text{C s}^{-1}$, much higher than graphene/MXene fabrics and PEDOT/PSS film shown in Table S3.”

Fig. R2-1 The devices of Joule heater assembled by $\text{Ti}_3\text{C}_2\text{T}_x$ and MEB films. (Fig.S33 in revised manuscript)

Fig. R2-2 Long-term Joule heating performance of MEB and $\text{Ti}_3\text{C}_2\text{T}_x$ films driven by an applied voltage of 3.0 V. (Fig. 5d in revised manuscript)

Table R2-1 Comparison of the steady temperature and rise rate for various Joule heating materials. (Table. S3 in revised manuscript)

Materials	Method	Steady temperature (°C)	Heating rate (°C/s)	Voltage (V)	Application	Reference
Graphene fabric	Spray-coating	162.6	8.4	10	Joule heater	15
Graphene-EC on Ag NW-based film	Spin-coating	99.8	3.4	12	Joule heater	16
PEDOT/PSS film	Deposition	100	1.6	12	Flexible transparent heater	17
Graphene	Electrochemical method	75	2.5	10	Ultrafast Electrothermal Heater	18
MXene/aramid nonwoven fabric	Spray coating	125	7.5	3	Electrothermal/photothermal conversion for wearable heater	19
		263	10	5		
MXene patten	Screen-printed	130.8	20	4	Joule heating	20
MXene/cellulose fabric	Dip coating	45	1	3	Healthcare and medical therapy	21
MXene freestanding film	Vacuum filtration	190	20	3	Joule heating	This work
MEB freestanding film	Vacuum filtration	198	20	3	Joule heating	This work

Reference

15. Tian, M. *et al.* Enhanced electrothermal efficiency of flexible graphene fabric Joule heaters with the aid of graphene oxide. *Mater. Lett.* **234**, 101-104 (2019).
16. Cao, M., Wang, M., Li, L., Qiu, H. & Yang, Z. Effect of graphene-EC on Ag NW-based transparent film heaters: optimizing the stability and heat dispersion of films. *ACS Appl. Mater. Interfaces* **10**, 1077-1083 (2018).
17. Gueye, M. N., Carella, A., Demadrille, R. & Simonato, J. P. All-polymeric flexible transparent heaters. *ACS Appl. Mater. Interfaces* **9**, 27250-27256 (2017).
18. Tian, S. *et al.* Electrochemical fabrication of high quality graphene in mixed

- electrolyte for ultrafast electrothermal heater. *Chem. Mater.* **29**, 6214–6219 (2017).
19. Wang, X. *et al.* A lightweight MXene-coated nonwoven fabric with excellent flame retardancy, EMI shielding, and electrothermal/photothermal conversion for wearable heater. *Chem. Eng. J.* **430**, (2022).
 20. Wu, H. *et al.* Aqueous MXene/Xanthan Gum hybrid inks for screen-printing electromagnetic shielding, Joule heater, and piezoresistive sensor. *Small* e2107087 (2022).
 21. Zhao, X. *et al.* Smart $\text{Ti}_3\text{C}_2\text{T}_x$ MXene fabric with fast humidity response and Joule heating for healthcare and medical therapy applications. *ACS Nano* **14**, 8793-8805 (2020).

Comment 2. *Is the extracted bentonite the only option? Are there any other group of materials that can offer these properties?*

Response: Based on our analysis, we think the extracted bentonite (EB) may be not the only option to obtain high-temperature resistant ability by compositing with $\text{Ti}_3\text{C}_2\text{T}_x$. EB is a layered material consisting of one Al-O octahedral sheet sandwiched by two Si tetrahedral nano-sheets with Na^+ ions to offset the charge imbalance. EB exhibits stronger binding ability with oxygen due to its un-saturated Si-3p orbitals. Upon interfacing with $\text{Ti}_3\text{C}_2\text{T}_x$ the saturated adsorption of oxygen on EB further inhibits more oxygen molecules to be absorbed on the surface of $\text{Ti}_3\text{C}_2\text{T}_x$ due to the weakened p-d orbital hybridization between adsorbed O_2 and $\text{Ti}_3\text{C}_2\text{T}_x$. This effect is induced by the $\text{Ti}_3\text{C}_2\text{T}_x$ /EB interface coupling. Therefore, we speculate that a layered material similar to EB with un-saturated p/d orbitals on surface would possess the high-temperature resistant trait, such as magadiite (*Clay Miner.* 2013, 48, 739-748), vermiculite (*Adv. Mater. Sci. Eng.* 2018, 309-316) and some layered alkali silicates (*J. Mater. Chem.* 2011, 21, 14336-14353; *Chem. Mater.* 2011, 23, 266–273), etc.

Comment 3. *Typically the oxidation of mxene flakes starting from the edge, but figure 1 the schematic figure seems to show oxidation occurring in between layers. Could the authors correct this image.*

Action: According to the reviewer’s suggestion, we have made a modification of Fig. 1 in main text of revised manuscript, as shown in Fig.R2-3.

Fig.R2-3 Schematic shows the oxidation process of pristine $Ti_3C_2T_x$ (starting from the edge) and suppressed oxidation of MEB under high temperature annealing with the presence of oxygen. (Fig.1 in revised manuscript)

Comment 4. In figureS3, its mentioned “Fig. S3 (a) Photograph of pristine $Ti_3C_2T_x$ and the MEB aqueous dispersions (with 10, 30, 50, 70, 90 wt.% EB which are the dispersions to prepare MEB-0.1, MEB-0.3, MEB-0.5, MEB-0.7, MEB-0.9, respectively). “there are five kinds of composite. but in the main manuscript, can you clarify which sample is being tested.

Response: In our manuscript, the sample named of MEB is short for MEB-0.5. We further revised the sentences for better understanding.

Action: We have modified the sentences in our revised manuscript, page 3:

“As shown in Fig. S3a, the mixed aqueous dispersions of $Ti_3C_2T_x/EB$ are aggregated when the mass ratio of EB is more than 50 wt.%. With the Zeta potential of -42 mV (Fig. S3b) close to sole $Ti_3C_2T_x$ or EB, the homogeneously mixed dispersion (Fig. S3c) with a EB mass ratio of 50 wt.% is chosen to fabricate $Ti_3C_2T_x/EB$ composite film for the following study and be denoted as MEB.”

Comment 5. How did u prepare the film in figure S4, in figure S4b, the SEM of MEB, which part is Mxene and which part is EB.

Response: Fig. R2-4 shows the preparing process of freestanding $Ti_3C_2T_x$ film and MEB (the same as EB film). In details, the MEB solutions (with 11 mg $Ti_3C_2T_x$ and 11 mg EB) were prepared by adding EB aqueous solution into the $Ti_3C_2T_x$ solution and continuous stirring for 2 h. The obtained homogeneous solution was filtrated by vacuum-assisted filtration and dried under vacuum at 60 °C for 24 h to obtain MEB (with the thickness of 11 μm). 11 mg freestanding pristine $Ti_3C_2T_x$ film (with the thickness of $\sim 5 \mu m$) and 22 mg freestanding pristine EB film (with the thickness of 11.2 μm) were fabricated by the same process.

As it is difficult to distinguish $Ti_3C_2T_x$ and EB in SEM, STEM-EDX of MEB was conducted to explain the distribution of EB and $Ti_3C_2T_x$ in the composite film. As shown in Fig. R2-5, layered and homogeneous distributions of Si and Ti elements indicate the stacking nature of EB and $Ti_3C_2T_x$.

Fig.R2-4 Schematic of the preparing process for freestanding $Ti_3C_2T_x$ film and MEB (the same as EB film).

Fig.R2-5 Cross-sectional STEM image of MEB with EDX mapping. (Fig. 2a in revised manuscript)

Comment 6. Which EB composition are you using in S2?

Response: In our work, sodium bentonite powder was chosen and be extracted, then delaminated to obtain EB nanosheets.

Comment 7. For figure 2c, What do u mean by thick chunk, which is it made of? And in the sem image, the top layers are showing very different deformation while the bottom part showing nothing, can u explain why?

Response: The cross-sectional SEM image in Fig. 2d shows the partial transformation from $Ti_3C_2T_x$ nanosheets to TiO_2 particles after annealing at 400 °C for 2h in air. The words of “thick chunk” are used to describe the aggregated TiO_2 particles.

Fig.R2-6 shows the schematic of the annealing process of our samples in this work. The sample was put into a quartz boat, then annealed in tube furnace under specific atmosphere (synthetic air, RH 90% argon or simulated atmospheric environment). During the annealing process, the bottom surface of sample may interact with less oxygen/water molecules than the upper surface due to the contact between bottom surface of sample and the quartz boat. This may result in less oxidation of the bottom surface than the top layer.

Fig.R2-6 Schematic of the annealing process of our samples in this work.

Comment 8. In page 9, humidity was also considered as a factor, but the test environment is 400 degree in argon with humidity 90%. But this is not a normal environment. The testing environment should be typical oxygen/nitrogen environment (air) with humidity within 100 degree.

Response: According to the reviewer’s suggestion, the high-temperature resistant of MEB annealing for 24 h at 100°C in RH 60% synthetic air with volume fractions of 21% O₂ and 79% N₂ (simulated atmospheric environment in practical scenario, sample name MEB-Atmos-100C-24) was investigated. As shown in Fig. R2-7a, the XRD patterns of MEB before and after treatment are almost the same without any SiO₂ and Al₂O₃ peaks appear, which indicates the antioxidation of MEB at 100°C in air. Furthermore, MEB-Atmos-100C-24 exhibits high THz EMI shielding performance of ~50 dB as shown in Fig. R2-7b.

Fig. R2-7 (a) XRD patterns of pristine MEB and MEB-Atmos-100C-24; (b) THz EMI shielding efficiency (SE) in 0.2-1.3 THz of MEB-Atmos-100C-24.

Comment 9. In figure S11, the xrd peaks seem to shift and broaden. It seems improper to make the deduction that the compositions disappear. The authors need to explain this phenomenon rather than stating that the the peaks are absent.

Response: We thank the reviewer to point this out and be sorry for our unclear description. Following the reviewer's suggestion, we have made an explanation:

In order to investigate the high-temperature resistant properties of EB, EB was annealed at 400 °C for 6h and 600°C for 2h in RH 90% synthetic air, respectively. The XRD pattern (the top pattern in Fig. R2-8) show SiO₂ and Al₂O₃ peaks in the oxidized EB. By contrast, the annealed EB exhibits the retention of (002) peaks without appearance of SiO₂ and Al₂O₃. For EB-RH 90% air-400C-6, the peak of (002) at $2\theta = 6.0^\circ$ become broaden, which is caused by partial collapsing of the stacking layer structure due to the removal of intercalated water and part of hydroxyl surface terminations. For EB-RH 90% air-600C-2, the (002) peak of EB shifts from $2\theta = 6.0^\circ$ to $\sim 8.0^\circ$, which indicates the decrease of interlayer space resulting from the removal of the terminations. Therefore, we speculate that EB is thermally stable during our annealing experiments with the presence of O₂ and H₂O.

Action: We added the sentences in the Supplementary Information:

“XRD patterns display the reduction of the interlayer (*d*-) spacing, characterized by the peak of (002) shifting from $2\theta = 6.0^\circ$ to $\sim 8.0^\circ$ after annealing at 600°C for 2 h. SiO₂ and Al₂O₃ are not observed after treatments, which reveals the thermal stability of EB in our annealing experiments with the presence of O₂ and H₂O.”

Fig. R2-8 XRD patterns of oxidized EB and EB after annealing under different conditions. (Fig. S11 in revised manuscript)

Comment 10. In figure S12, EB 400C-6 h and 600C-2h were compared. Temperature and duration are both changing, so it is hard to compare.

Response: Thanks for the reviewer's comments. To avoid misunderstandings, we have made an explanation and modification:

In order to investigate the high-temperature resistant properties of EB, EB was annealed at 400 °C for 6h and 600°C for 2h in RH 90% synthetic air respectively, which are the harshest treatments for MEB in our experiment. The structures of the annealing samples (Fig. R2-10 and R2-11) were compared with the pristine sample (Fig. R2-9), respectively. Therefore, we have divided the original Fig. S12 into two figures.

Action: We have made a modification of the figures in Supplementary Information:

Fig. R2-9 Photographs of EB film, and the corresponding cross-sectional SEM image. (Fig. S4c in revised manuscript)

Fig. R2-10 SEM images of the cross-section of EB-RH 90% air-400C-6. (Fig. S12 in revised manuscript)

Fig. R2-11 SEM images of the cross-section EB-RH 90% air-600C-2. (Fig. S13 in revised manuscript)

Reviewer #3

Comment 1. *The authors have revised and restructured the manuscript by addressing some of reviewers' comments. Compared to the previous version, the authors, in fact, have not provided new information on the materials and their behaviour as was required by the reviewers, but only further explanation of existing results.*

Specifically,

- Oxidation issue: I am satisfied with the response provided by the authors on the spectroscopic findings suggesting that oxidation is not completely prevented in MEB. EB is supposed to suppress (I would say "delay") the diffusion of O₂. However, the "further comment" I was expecting in the main text has not been added (even in the SI!). In this regard, clear/ direct information on the water/oxygen barrier properties (i.e. permeability!) of the system was not provided; therefore I am not fully satisfied with the response of the authors to comment 3.

Response: Thanks for the reviewer's comments. We are terribly sorry for misunderstanding the reviewer's suggestion at first. The permeabilities of water and oxygen for Ti₃C₂T_x and MEB membranes has been conducted, as shown in Table R3-1. The O₂/H₂O permeability of Ti₃C₂T_x membrane is much higher than MEB, which reveals the better barrier property of MEB possibly due to the compact layer structure as we proposed.

Action: We have made a modification, at Page 9 and 14:

“This is further evidenced by the higher permeability of O₂/H₂O of Ti₃C₂T_x than MEB as shown in Table S1, which reveals the better barrier property of MEB possibly benefitted by the compact layer structure. As a result, the MEB composite could suppress the diffusion of O₂ to some extent, lowering the amount of O₂/H₂O around Ti₃C₂T_x.”

“The O₂ permeability was measured using a permeability tester (OX-TRAN 2/12R, MOCON Co., USA). The water vapor transmission rate was measured using a moisture permeability testing apparatus (C360, Jinan Languang Electromechanical Technology Co.).”

Table R3-1. The permeabilities of water and oxygen for $\text{Ti}_3\text{C}_2\text{T}_x$ and MEB membranes. (Table S1 in revised manuscript)

Sample	Atmosphere	Permeability
$\text{Ti}_3\text{C}_2\text{T}_x$	O ₂	244713 cm ³ /(m ² ·day)
	H ₂ O	296 g/(m ² ·day)
MEB	O ₂	128856 cm ³ /(m ² ·day)
	H ₂ O	241 g/(m ² ·day)

Comment 2. - *Mechanical integrity of the membranes before and after the treatment: as mentioned in the previous round of revision, optical images of bent or deformed membranes are not sufficient in order to assess the mechanical integrity! Proper mechanical characterization results (e.g. stress/strain curves) are required for completeness and they are not provided in the revised version of the manuscript.*

Response: According to the reviewer's suggestion, we added tensile stress-strain curves of the membranes before and after the treatment, as shown in Fig.R3-1. The pristine $\text{Ti}_3\text{C}_2\text{T}_x$ and MEB membranes present 26 MPa and 23.5MPa (influenced by EB) of tensile stress, respectively. After annealing at 400 °C for 2h in air, MEB exhibits 10.6% decrease of the tensile stress and 1.0% decrease of strain. By contrast, $\text{Ti}_3\text{C}_2\text{T}_x$ exhibits 57.7% decrease of the tensile stress and 5.5% decrease of strain, which is caused by the degradation of $\text{Ti}_3\text{C}_2\text{T}_x$ in annealing treatment. The results reveal that EB can delay the degradation of $\text{Ti}_3\text{C}_2\text{T}_x$ in MEB at high temperature in air.

Action: We made a modification of Figure 2c and added the analysis at Page 4 and 14:

“As shown in Fig. 2b, MEB-Air-400C-2 reserves the flexibility of fresh MEB and keeps its integrity after ultrasonication for 1 min in DI water (Fig. S5b). By contrast, $\text{Ti}_3\text{C}_2\text{T}_x$ -Air-400C-2 tends to be fragile (Fig. 2b and Fig. S5a). According to the tensile stress-strain test (Fig. 2c and Fig. S6), MEB-Air-400C-2 exhibits 10.6% decrease of the tensile stress and 1.0% decrease of strain. By contrast, $\text{Ti}_3\text{C}_2\text{T}_x$ -Air-400C-2 exhibits 57.7% decrease of the tensile stress and 5.5% decrease of strain, which is caused by the degradation of $\text{Ti}_3\text{C}_2\text{T}_x$ in high-temperature treatment.”

“Mechanical tests of films were performed on a dynamic mechanical analyzer WDW-10D and DS2-50N-XB (Baichuan instruments Co.). The $Ti_3C_2T_x$ film samples were cut into 10 mm × 30 mm strips using a steel razor blade. Uniaxial tensile tests were performed at a strain rate of 5mm/min. The sample thickness was estimated by the helical micrometer.”

Fig. R3-1 The tensile stress-strain curves of MEB-RT, MEB-Air-400C-2, $Ti_3C_2T_x$ -RT, $Ti_3C_2T_x$ -Air-400C-2, and EB. (Fig.2c and Fig.S6 in revised manuscript)

Comment 3. -EMI shielding: I am now satisfied with the analysis of the data based on the specific SE to compare with SOTA materials; however, theoretical predictions of the different shielding contributions (that were clearly required in my comment) was not provided in the revised manuscript. Without some theoretical predictions, the qualitative explanation of the shielding mechanism that has been added is not satisfactory.

Response: We agree to the reviewer’s suggestion that analyzing different shielding contributions is important for fundamental understanding, even though our work is focusing more on material properties. Accordingly, we calculated the different shielding contributions, including reflection and absorption. Specially, the contribution of multiple reflection is included in the absorption. The results are shown in Fig. R3-2.

Action: In the Supplementary Information, we have added Fig. S31 and made an explanation:

“It can be clearly observed that the transmitted THz signals of $\text{Ti}_3\text{C}_2\text{T}_x$ -RT and MEB-RT are close to 0% (Fig. S31a), which is due to the excellent THz shielding performance of $\text{Ti}_3\text{C}_2\text{T}_x$ with the domination of THz reflection (Fig. S31b, e, f). After annealing, MEB-Atmos-400C-6 have a good shielding capacity retention, with a contribution of ~65% THz reflection and ~35% THz absorption (Fig. S31b, f). By contrast, $\text{Ti}_3\text{C}_2\text{T}_x$ -Atmos-400C-6 performs high transmittance (~58%) and ultralow reflection (approach to zero) owing to its degradation. With the annealing temperature increasing, the transmissions of THz waves through MEB-Atmos-500C-2h and MEB-Atmos-600C-2h remain at ~ 0%, with the decrease of reflection and increase of absorption. On the contrary, $\text{Ti}_3\text{C}_2\text{T}_x$ -Atmos-600C-2h exhibits ~0 dB THz EMI shielding efficiency, because of the oxidation of $\text{Ti}_3\text{C}_2\text{T}_x$. All the results suggest that the introducing of EB can suppress the oxidation-induced deterioration of $\text{Ti}_3\text{C}_2\text{T}_x$, therefore making MEB being promising for THz shielding at high temperature in oxidizing environment for a long operation time.”

Fig. R3-2 High-temperature resistant THz shielding property. THz transmittance (a) and reflection (b) in 0.2-1.3 THz of $Ti_3C_2T_x$ -RT, MEB-RT, $Ti_3C_2T_x$ -Atmos-400C-6, MEB-Atmos-400C-6. THz transmittance (c) and reflection (d) in 0.2-1.3 THz of $Ti_3C_2T_x$ -Atmos-500C, 600C-2 and MEB-Atmos-500C, 600C-2. The average THz transmittance, reflection and absorption of $Ti_3C_2T_x$ (e) and MEB (f) before and after treatments. (Fig. S31 in revised manuscript)